# Paf1 and Ctr9 subcomplex formation is essential for Paf1 complex assembly and functional regulation

Ying Xie[1], Minying Zheng[1], Xinlei Chu[2], Yue Chen[1], Huisha Xu[1], Jiawei Wang[3], Hao Zhou[1] & Jiafu Long [1]

The evolutionarily conserved multifunctional polymerase-associated factor 1 (Paf1) complex (Paf1C), which is composed of at least five subunits (Paf1, Leo1, Ctr9, Cdc73, and Rtf1), plays vital roles in gene regulation and has connections to development and human diseases. Here, we report two structures of each of the human and yeast Ctr9/Paf1 subcomplexes, which assemble into heterodimers with very similar conformations, revealing an interface between the tetratricopeptide repeat module in Ctr9 and Paf1. The structure of the Ctr9/Paf1 subcomplex may provide mechanistic explanations for disease-associated mutations in human PAF1 and CTR9. Our study reveals that the formation of the Ctr9/Paf1 heterodimer is required for the assembly of yeast Paf1C, and is essential for yeast viability. In addition, disruption of the interaction between Paf1 and Ctr9 greatly affects the level of histone H3 methylation in vivo. Collectively, our results shed light on Paf1C assembly and functional regulation.

[1] State Key Laboratory of Medicinal Chemical Biology, Tianjin Key Laboratory of Protein Science, and College of Life Sciences, Nankai University, 94 Weijin Road, Tianjin 300071, China. [2] Department of Epidemiology and Biostatistics, Tianjin Medical University Cancer Institute and Hospital, Tianjin 300060, China. [3] State Key Laboratory of Membrane Biology, School of Life Sciences, Tsinghua University, Beijing 100084, China. These authors contributed equally: Ying Xie, Minying Zheng. Correspondence and requests for materials should be addressed to H.Z. (email: haozhou@nankai.edu.cn) or to J.L. (email: jflong@nankai.edu.cn)

The highly conserved eukaryotic multifunctional polymerase-associated factor 1 (Paf1) complex (Paf1C) was originally identified in budding yeast over 2 decades ago[1], and robust data suggested that it plays a vital role in transcriptional elongation and chromatin modifications[2,3]. Yeast Paf1C comprises subunits Paf1, Leo1, Ctr9, Cdc73, and Rtf1[4], while the human homolog PAF1C (human PAF1C and each subunit are written in all caps to distinguish yeast Paf1C and subunits, hereafter) contains an additional subunit, SKI8, which belongs to the SKI complex known for RNA quality control[5]. PAF1C is recruited to the transcription machinery by CTR9[6] or CDC73[7] binding to RNA polymerase II (pol II). In addition to its interactions with pol II, Paf1C has extensive genetic and physical links to yeast elongation factors Spt4/Spt5, Spt16/Pob3, and Dst1 (corresponding to human DSIF, FACT, and TFIIS/SII, respectively)[4,6,8], allowing it to regulate transcription elongation procession.

Paf1C or PAF1C is also involved in transcription-coupled histone modifications. Paf1C or PAF1C is required to promote H2B ubiquitination, as well as H3 methylation at K4, K36, and K79 in yeast, flies, and humans[5,9–13]. Paf1C or PAF1C participates in mRNA 3'-end processing[14–18], and nuclear export of transcripts[19], thereby linking transcription and posttranscriptional regulations. Paf1C or PAF1C participates in such other cellular functions as cell cycle regulation[20], stem cell pluripotency maintenance[21], signal transduction[22], DNA repair[23,24], small-RNA-mediated gene silencing[25] or activation[26], and the regulation of promoter-proximal pausing of pol II[27,28], or the release of paused pol II[29–31], depending on context.

PAF1C has also been implicated in tumorigenesis[3]. Thus, subunits of PAF1C could act as oncogenes or tumor suppressors in a context-dependent manner. For example, the overexpression of PAF1 promotes pancreatic cancer formation[32], and regulates self-renewal and drug resistance of pancreatic and ovarian cancer stem cells[33]. In contrast, germline mutations in CTR9 predispose to Wilms tumor[34], suggesting that CTR9 may act as a tumor suppressor. Furthermore, Wei Xu and coworkers discovered that CTR9 is a central regulator of estrogen signaling that drives ERα⁺ breast tumorigenesis[35]. CDC73, encoded by *HRPT2*, is a tumor suppressor that is mutated in the germline of hyperparathyroidism–jaw tumor syndrome (OMIM 145001)[7,36]. Interestingly, CDC73 is amplified in liver carcinoma and breast cancer[37].

The structures of Ras-like domain of Cdc73[38,39] and the Plus3 domain of RTF1[40–42] provide the structural basis for Paf1 complex chromatin association. Recently, the crystal structure of the N-terminal domain of CDC73 has been resolved, which may provide the molecular mechanisms of hyperparathyroidism–jaw tumor mutants[43]. The structure of histone modification domain (HMD) of Rtf1 was reported and the HMD was shown to stimulate H2B ubiquitylation through interaction with Rad6[44]. However, there is no atomic structure information has been reported about CTR9, which contains multimotifs for protein–protein interaction (Fig. 1a).

To date, structural information on yeast Paf1C or human PAF1C assembly is limited, though an architecture of yeast Paf1C with a local resolution of ~10–20 Å was obtained recently[45]. In a previous study, we determined the crystal structure of the human PAF1/LEO1 heterodimer and showed that CTR9 acts as a key scaffold protein during PAF1C assembly[46]. In this study, we determine the atomic structures of both the human PAF1/CTR9 and yeast Paf1/Ctr9 heterodimers, both with a resolution of ~2.53 Å. Our results clearly reveal that the heterodimer of PAF1/CTR9 or Paf1/Ctr9 is the core component and is important for human PAF1C or yeast Paf1C assembly, yeast viability, and Paf1C-mediated histone modifications. Our study suggests that the PAF1/CTR9 or Paf1/Ctr9 subcomplex-mediated holocomplex assembly and functional regulation may be a general mechanism for complexes from other eukaryotic species.

## Results

**The interaction between human CTR9 and PAF1.** Structural and biochemical data reveal that LEO1 binds to human PAF1C through PAF1 and that the CTR9 subunit is the key scaffold protein in assembling PAF1C[46]. Protein sequence analysis showed that both human CTR9 (Fig. 1a) and yeast Ctr9 (see below) possessed multiple tetratricopeptide repeat (TPR) motifs. We confirmed that the human CTR9, PAF1, and LEO1 (herein named CTR9/PAF1/LEO1, "/" denotes protein complexes with separate chains, similar nomenclature hereafter) subunits interacted by showing that three thioredoxin (Trx)-tagged fusion proteins [aa 1–284, Trx-CTR9$^{(1-284)}$, Trx-PAF1, and Trx-LEO1] eluted from the size-exclusion column as a tripartite complex (Supplementary Fig. 1a). Next, we sought to test the heteromeric complex formation between PAF1 and CTR9$^{(1-284)}$. We found that a conserved N-terminal fragment of PAF1 (aa 57–266, PAF1$^{(57-266)}$) formed a stable dimeric complex with CTR9$^{(1-284)}$ (Supplementary Fig. 1b). Using a similar truncation-based approach, we mapped the minimal PAF1-binding region, including TPR1–5 of CTR9, to a 249-residue fragment (aa 1–249, CTR9$^{(1-249)}$) and the minimal CTR9-binding region of PAF1 to a 60-residue fragment (aa 57–116, PAF1$^{(57-116)}$) (Fig. 1a and Supplementary Fig. 1c–e). It was noted that CTR9$^{(1-249)}$ and PAF1$^{(57-116)}$ assembled into a complex since their co-elution in the analytical size-exclusion column (red line and sodium dodecyl sulfate-polyacrylamide gel electrophoresis (SDS-PAGE) inset in Supplementary Fig. 1e). Analytical ultracentrifugation further confirmed that the CTR9$^{(1-249)}$/PAF1$^{(57-116)}$ complex formed a stoichiometry of a 1:1 heterodimer with a molecular mass of ~34 kDa (red line in Supplementary Fig. 1f). More, to further validate the role of PAF1$^{(57-116)}$ in CTR9/PAF1 complex formation, the N-terminal 116 amino acids of PAF1 were deleted in mutant [referred to as PAF1(N116Δ)]. Notably, green fluorescent protein (GFP)-tagged PAF1(N116Δ) [GFP-PAF1(N116Δ)] mutant did not co-immunoprecipitate with Myc-tagged CTR9 (Myc-CTR9) thoroughly (lane 3 in Supplementary Fig. 1g), whereas GFP-PAF1 specifically formed a complex with Myc-CTR9 (lane 2 in Supplementary Fig. 1g) in a co-immunoprecipitation (co-IP) assay, indicating that the N-terminal fragment including amino acids 57–116 of PAF1 is essential for the interaction between full-length CTR9 and PAF1. Given these results, we conclude that the CTR9/PAF1 subcomplex forms a heterodimer through interaction between fragments CTR9$^{(1-249)}$ and PAF1$^{(57-116)}$.

**Crystal structure of the CTR9 and PAF1 heterodimer.** To understand how CTR9 and PAF1 bind to each other, we attempted to determine the crystal structure of the CTR9$^{(1-249)}$/PAF1$^{(57-116)}$ heterodimer using two separate chains; these experiments were unsuccessful. However, we succeeded in obtaining crystals of the single polypeptide created by the fusion of CTR9$^{(1-249)}$ to the N-terminus of PAF1$^{(57-116)}$ with a tobacco etch virus (TEV)-cleavable segment. This technique has been used previously to determine the PAF1/LEO1 heterodimer structure[46]. The purified single-chain fusion protein of CTR9$^{(1-249)}$ and PAF1$^{(57-116)}$ (herein named CTR9$^{(1-249)}$–PAF1$^{(57-116)}$, "-" denotes proteins in a single-chain fusion, similar nomenclature hereafter) eluted as a single peak from an analytical size-exclusion column (black line in Supplementary Fig. 1e) and assembled into a heterodimer with a molecular mass of ~35 kDa from the sedimentation velocity (SV) experiment (black line in Supplementary Fig. 1f). The crystal structure of CTR9$^{(1-249)}$–PAF1$^{(57-116)}$ was determined at a resolution of 2.53 Å and with one molecule per asymmetric unit (Supplementary Table 1).

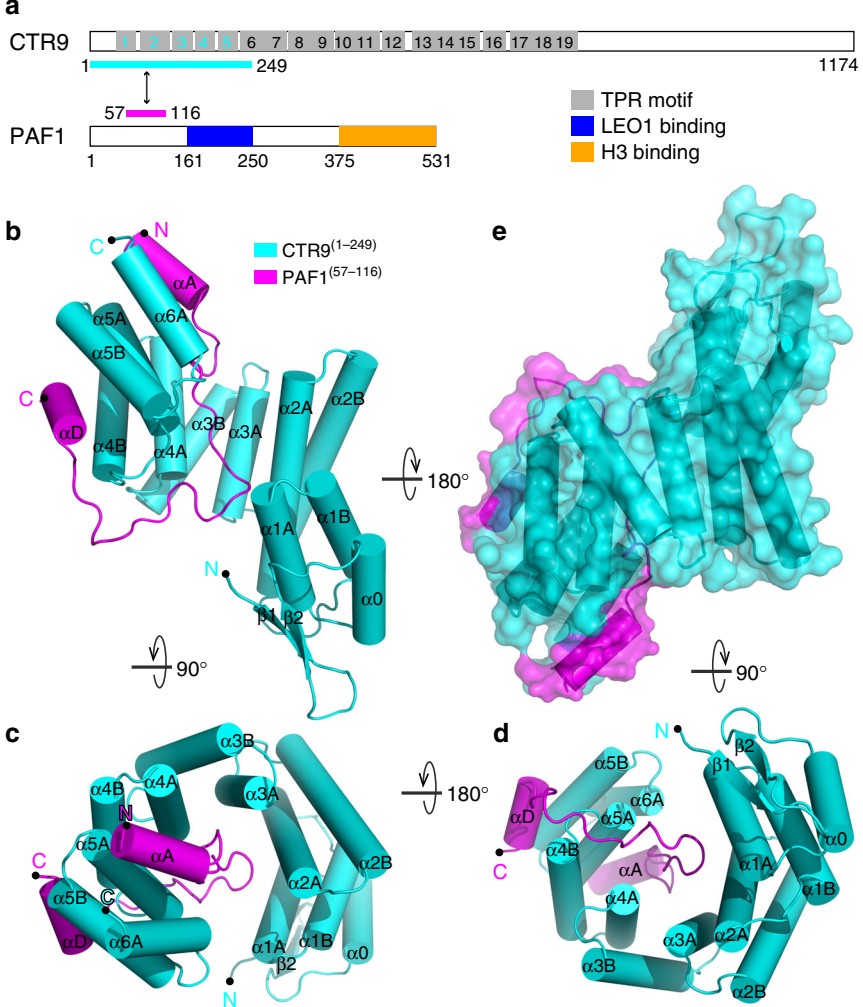

**Fig. 1** Crystal structure of the human CTR9/PAF1 heterodimer. **a** Schematic representation of full-length CTR9 and PAF1. A LEO1-interacting region (blue) and a histone H3-interacting region (orange) are shown in PAF1. The predicted 19 TPR motifs (gray) were defined using TPRpred (https://toolkit. tuebingen.mpg.de/#/tools/tprpred). The protein fragments of the CTR9$^{(1-249)}$/PAF1$^{(57-116)}$ complex used for structural determination are indicated by a two-way arrow and are colored cyan and magenta, respectively. **b** Ribbon diagram representation of the CTR9$^{(1-249)}$ (cyan)/PAF1$^{(57-116)}$ (magenta) complex viewed from the side. The N- and C-termini of the two proteins are labeled. Cylinder representation of the CTR9$^{(1-249)}$/PAF1$^{(57-116)}$ complex structure viewed from the top (**c**) or the bottom (**d**). **e** Surface representation of the CTR9$^{(1-249)}$/PAF1$^{(57-116)}$ complex with its orientation corresponding to the other side from that shown in (**b**)

In the final model, all residues were clearly visible, except residues 1–2 and 246–249 of CTR9$^{(1-249)}$ and residues 57–66 and 110–116 of PAF1$^{(57-116)}$, presumably owing to conformation flexibility (Fig. 1b–e, Fig. 2e, and Supplementary Figs. 2 and 3a). Interestingly, we noted that the entire artificial TEV-cleavable linker was missing in the final model (Supplementary Fig. 3b). This was seemingly caused by the existence of multiple conformers of the linker in the current structure because SDS-PAGE analysis of dissolved CTR9$^{(1-249)}$–PAF1$^{(57-116)}$ crystals demonstrated that the molecules were intact (Supplementary Fig. 3c). These data suggested that the artificial linker did not alter the global structure of the CTR9$^{(1-249)}$–PAF1$^{(57-116)}$ and may stabilize the N-terminus of PAF1$^{(57-116)}$ to help complex proteins crystallization.

The structure of CTR9$^{(1-249)}$ was composed of 12 α-helices and two β-strands at the N-terminus (Fig. 1b–d and supplementary Fig. 2). The helices α1A to α5B contained five sequential helix-turn-helix repeats that exhibited structural similarity to TPR (Fig. 1b–d and Supplementary Fig. 4a). All five of these TPR motifs, named TPR1 (α1A and α1B), TPR2 (α2A and α2B), TPR3 (α3A and α3B), TPR4 (α4A and α4B), and TPR5 (α5A and α5B),

were conserved with the canonical TPR motif, except TPR2, which had both a longer A helix and B helix and was distinct from the canonical TPR motif containing 34 amino acids and other noncanonical TPR motifs (Supplementary Fig. 4b and refs. 47 and 48). Like multiple other TPR motifs[49–53], the five TPR units of CTR9$^{(1-249)}$ were arranged in parallel with adjacent α-helices to each other to create a right-handed superhelix (Fig. 1b–d). It was also noted that there was an N-terminal cap composed of the antiparallel β-sheet (β1 and β2) and the α0 helix and a C-terminal cap composed of the α6A helix, which interacted with TPR1 and TPR5, respectively (Fig. 1b–d). The PAF1$^{(57-116)}$ structure, which was composed of a long loop flanked by two α helices (αA and αD) in the N- and C-terminus, respectively, adopted a hook-shaped conformation occupying the entire concave channel formed by subdomains of TPR1–3 and TPR4–5 and the convex surface formed by TPR4–5 (Fig. 1b–d).

**CTR9 and PAF1 interactions in the heterodimer.** The CTR9/ PAF1 interface could be divided into three regions according to

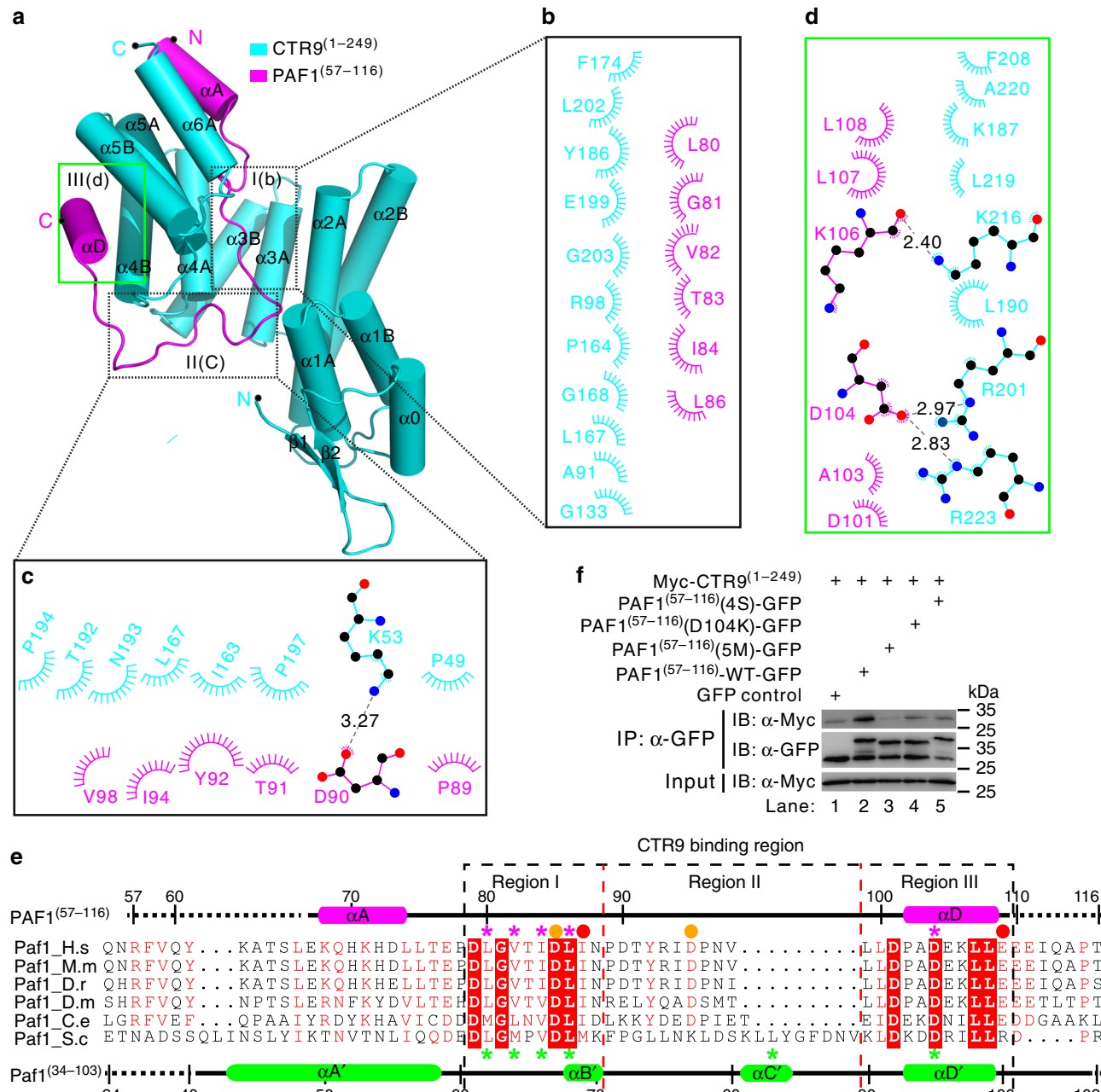

**Fig. 2** The interaction interface of the human CTR9/PAF1 heterodimer. **a–e** The CTR9$^{(1-249)}$/PAF1$^{(57-116)}$ interface is divided into three regions corresponding to the CTR9 TPR1–5 concave channel/N-terminal loop of PAF1 (**b**, **e**), the side of the TPR1–5/C-terminal loop of PAF1 (**c**, **e**), and the convex surface of the TPR4–5/αD helix of PAF1 (**d**, **e**). The interaction details between CTR9 and PAF1 in the three regions are shown in (**b**) to (**d**). Charge–charge or hydrogen-bonding and hydrophobic interactions are shown as gray dotted lines and spoked arcs, respectively. **e** Structural-based sequence alignment of the CTR9-binding fragments of PAF1 in various species. In this alignment, the secondary structures of human PAF1 and yeast Paf1 are shown at the top and bottom, respectively, according to the crystal structures of CTR9$^{(1-249)}$–PAF1$^{(57-116)}$ and Ctr9$^{(1-313)}$–Paf1$^{(34-103)}$, and conserved residues are shaded in red. The highly conserved residues, which are mutated in PAF1$^{(57-116)}$(4S), PAF1$^{(57-116)}$(D104K), or PAF1$^{(57-116)}$(5M) and the Paf1(4S), Paf1(L83S) or Paf1 (D95K) construct, are indicated with magenta and green asterisks, respectively. The disease-associated amino acid substitutions I87M or E109K (category 1 shown in Fig. 3a) and D85N or D95Y (category 2 shown in Fig. 3a) are indicated with red and orange spheres, respectively. Three regions of CTR9-binding in PAF1 are indicated with dotted boxes. Species abbreviations: H.s, *Homo sapiens*; M.m, *Mus musculus*; D.r, *Danio rerio*; D.m, *Drosophila melanogaster*; C.e, *Caenorhabditis elegans*; S.c, *Saccharomyces cerevisiae*. **f** Co-IP experiments testing the interaction between CTR9$^{(1-249)}$ and PAF1$^{(57-116)}$ wild-type (WT) or mutants. The PAF1$^{(57-116)}$(4S) mutant contains four amino acid substitutions L80S, V82S, I84S, and L86S. The PAF1$^{(57-116)}$(5M) mutant contains five amino acid substitutions L80S, V82S, I84S, L86S, and D104K. Myc was tagged to the N-terminal of CTR9$^{(1-249)}$ and GFP was tagged to the C-terminal of PAF1$^{(57-116)}$ WT or mutant. Extracts were prepared from HEK293T cells transfected with various combinations of plasmids, as indicated. The bottom panel shows 3% of the Myc fusion proteins for each IP. Uncropped blots are shown in Supplementary Fig. 8

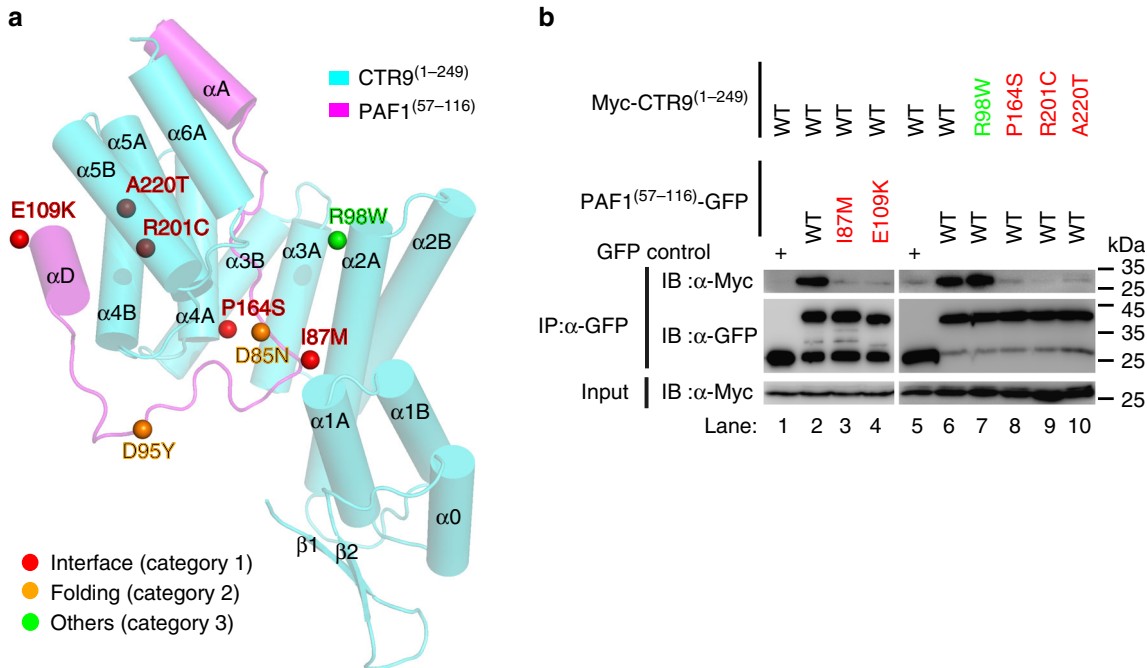

**Fig. 3** Disease-associated mutations affect the interaction between CTR9 and PAF1. **a** Disease-associated mutations in the CTR9$^{(1-129)}$/PAF1$^{(57-116)}$ complex. For clarify, only five missense-mutation sites in category 1 (interface), two sites in category 2 (folding) of PAF1, and one site (R98W) in category 3 (others) of CTR9 are highlighted with spheres and colored in red, orange, and green, respectively. The full lists of disease-associated mutations in CTR9 and PAF1 are summarized in Supplementary Tables 2 and 3, respectively. **b** The interaction sites between CTR9$^{(1-249)}$ and PAF1$^{(57-116)}$ containing various disease-associated mutations were evaluated using a co-IP strategy. Myc was tagged to the N-terminal of CTR9$^{(1-249)}$ WT or mutant and GFP was tagged to the C-terminal of PAF1$^{(57-116)}$ WT or mutant. Extracts were prepared from HEK293T cells transfected with various combinations of plasmids, as indicated. The bottom panel shows 3% of the Myc fusion proteins for each IP. Uncropped blots are shown in Supplementary Fig. 8

the heterodimer structure (Fig. 2a, e): region I: the N-terminal half-loop of PAF1$^{(57-116)}$ and the concave channel of TPR1–5 (Fig. 2b); region II: the C-terminal half-loop of PAF1$^{(57-116)}$ and the side of TPR1–5 (Fig. 2c); and Region III: the αD helix at the C-terminus of PAF1$^{(57-116)}$ and the convex surface of TPR4–5 (Fig. 2d). Region I was mainly maintained through four conserved hydrophobic amino acids (L80, V82, I84, and L86) in PAF1 and corresponding amino acids in CTR9 by hydrophobic interaction (Fig 2b, e, and Supplementary Fig. 5a). Region II was mainly mediated by two types of interactions between amino acid pairs from PAF1 and CTR9, respectively: hydrogen bond in D90/K53 pair and hydrophobic interactions (such as P89, T91, T92, I94, and V98 in PAF1 and corresponding amino acids in CTR9) (Fig. 2c and Supplementary Fig. 5b). The binding of Region III was largely mediated through hydrogen bonds by several pairs of amino acids between PAF1 and CTR9: D104/R201, D104/R223, and K106/K216, which formed an extensive network of hydrogen bonds (Fig. 2d and Supplementary Fig. 5c).

We performed a series of mutagenesis studies to validate the interactions observed in the structure of the CTR9$^{(1-249)}$–PAF1$^{(57-116)}$ complex. To this end, four conserved hydrophobic residues from region I (L80, V82, I84, and L86) were substituted with serine in PAF1$^{(57-116)}$ [referred to as PAF1$^{(57-116)}$(4S)], residue D104 from region III, which is conserved from yeast to human, was mutated to lysine in PAF1$^{(57-116)}$ [referred to as PAF1$^{(57-116)}$(D104K)], and all five amino acids substitutions L80S, V82S, I84S, L86S, and D104K were included in PAF1$^{(57-116)}$ [referred to as PAF1$^{(57-116)}$(5 M)]. Consistent with the structure-based prediction, the purified PAF1$^{(57-116)}$(5 M) could not form a stable complex with CTR9$^{(1-249)}$ in the analytical size-exclusion column (dashed line in Supplementary Fig. 6a and SDS-PAGE in

Supplementary Fig. 6a2). Interestingly, it was noted that CTR9$^{(1-249)}$ lacking of interaction with the PAF1$^{(57-116)}$(5 M) mutant or without PAF1$^{(57-116)}$ is aggregated in size-exclusion column (dotted line and blue line comparing to black line in Supplementary Fig. 6a and SDS-PAGEs in Supplementary Fig. 6a2 and a3 comparing to a1), indicating that CTR9$^{(1-249)}$/PAF1$^{(57-116)}$ heterodimer formation is important for maintaining the conformation of CTR9$^{(1-249)}$. More, consistent with this result, the C-terminal GFP-tagged mutants PAF1$^{(57-116)}$(5M), PAF1$^{(57-116)}$(D104K), and PAF1$^{(57-116)}$(4S) [PAF1$^{(57-116)}$(5M)-GFP, PAF1$^{(57-116)}$(D104K)-GFP, and PAF1$^{(57-116)}$(4S)-GFP] did not co-immunoprecipitate with Myc-tagged CTR9$^{(1-249)}$ (Myc-CTR9$^{(1-249)}$) (lanes 3–5 in Fig. 2f), whereas PAF1$^{(57-116)}$-GFP WT specifically formed a complex with Myc-CTR9$^{(1-249)}$ (lane 2 in Fig. 2f) in a co-IP assay. Additionally, the structure of CTR9$^{(1-249)}$–PAF1$^{(57-116)}$ indicated that the αA of PAF1$^{(57-116)}$ may be not important for CTR9$^{(1-249)}$/PAF1$^{(57-116)}$ heterodimer formation (Fig. 2). To further validate this observation, a truncated fragment of PAF1 (aa 75–116, PAF1$^{(75-116)}$) was designed to delete the N-terminal αA of PAF1$^{(57-116)}$. As expected, size-exclusion chromatography and SDS-PAGE analysis showed that PAF1$^{(75-116)}$ formed a complex with CTR9$^{(1-249)}$ (cyan line and SDS-PAGE inset in Supplementary Fig. 1h). However, a temperature-dependent denaturation assay demonstrated that the CTR9$^{(1-249)}$/PAF1$^{(75-116)}$ complex is less stable than the CTR9$^{(1-249)}$/PAF1$^{(57-116)}$ complex (cyan line comparing to red line and insert in Supplementary Fig. 1i), indicating that the αA of PAF1$^{(57-116)}$ may contribute to the stability of the CTR9$^{(1-249)}$/PAF1$^{(57-116)}$ complex. Collectively, these results confirm the interaction mode revealed by the structure of CTR9$^{(1-249)}$–PAF1$^{(57-116)}$.

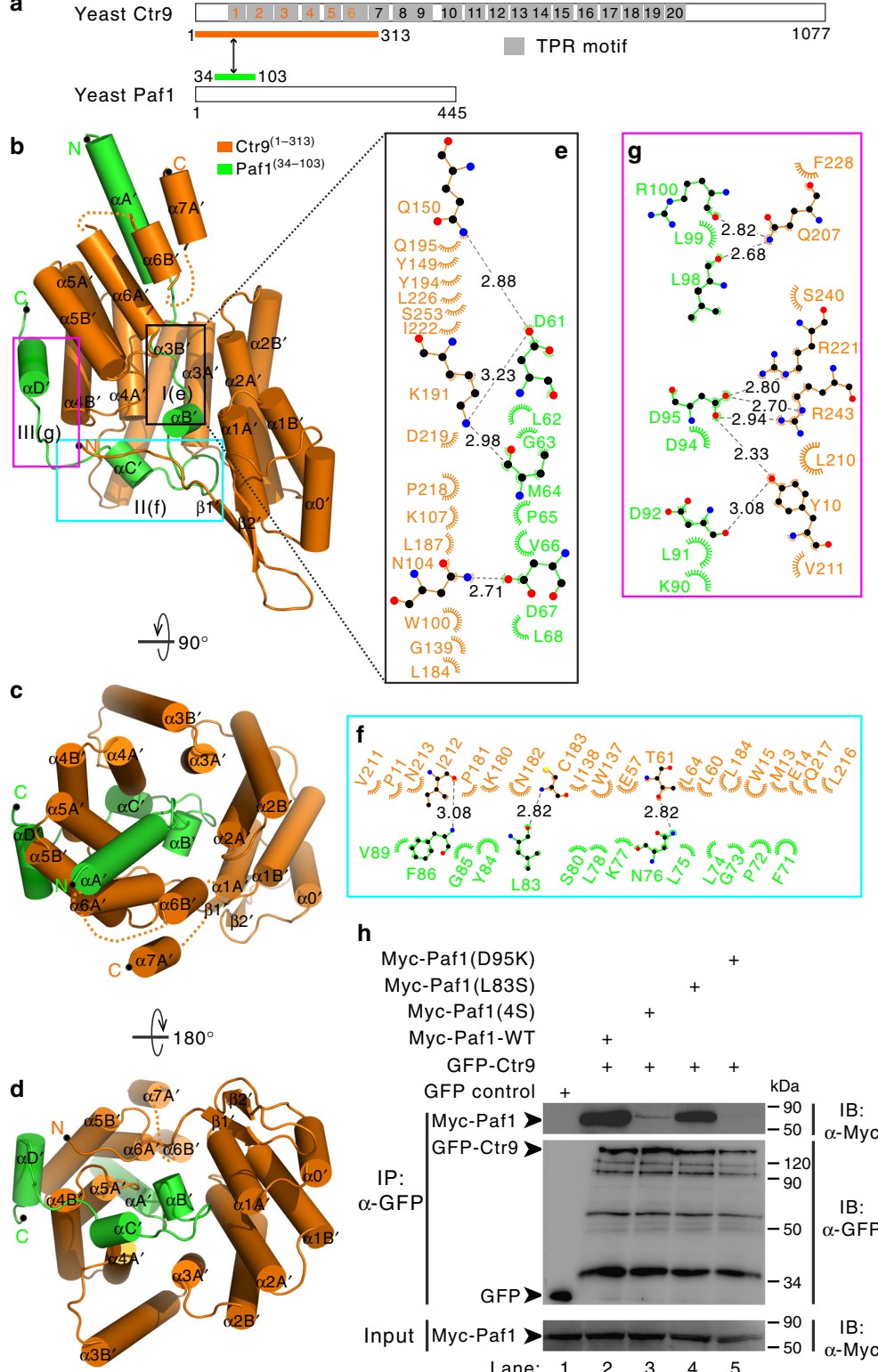

**Disease-associated mutations affect CTR9 binding to PAF1.** The deletion of the *CTR9* gene locus with *PAF1* locus amplification may synergistically lead to initiation and progression of pancreatic cancer[37]. A total of 38 and 17 missense mutations were located in 38 residues of CTR9[(1–249)] and 12 residues of PAF1[(57–116)] (Supplementary Tables 2 and 3), respectively, based on the data extracted from the COSMIC database (http://cancer.sanger.ac.uk/cosmic). We then mapped the 55 mutations to the modeled

structure and divided these mutations into three categories (named as interface, folding, and others, respectively, only mutations used in this study were shown in Fig. 3a and the full list of mutations was shown in Supplementary Table 2 or 3). The most interesting category (interface), which was directly relevant to this study, contained five mutations (P164S, R201C, and A220T of CTR9[(1–249)] and I87M and E109K of PAF1[(57–116)]) located at the interface of the CTR9[(1–249)]–PAF1[(57–116)] heterodimer. Category 2

**Fig. 4** Overall structure of the yeast Ctr9/Paf1 heterodimer. **a** Schematic representation of full-length Ctr9 and Paf1. The 20 predicted TPR motifs (gray) were defined using TPRpred. The protein fragments of the Ctr9$^{(1-313)}$/Paf1$^{(34-103)}$ complex used for structural determination are indicated by a two-way arrow and are colored orange and green, respectively. **b** Cylinder representation of the Ctr9$^{(1-313)}$ (orange)/Paf1$^{(34-103)}$ (green) complex viewed from the side. The N- and C-termini of the two proteins are labeled. The Ctr9$^{(1-313)}$/Paf1$^{(34-103)}$ complex structure viewed from the top (**c**) or the bottom (**d**). **e–g** The Ctr9$^{(1-313)}$/Paf1$^{(34-103)}$ interface is divided into three regions corresponding to the Ctr9 TPR1–5 concave channel/N-terminal loop of Paf1 (**e** and Fig. 2e), the side of the TPR1–5/C-terminal loop of Paf1 (**f** and Fig. 2e), and the convex surface of the TPR4–5/αD' helix of Paf1 (**g** and Fig. 2e). The interaction details between Ctr9 and Paf1 in the three regions are shown in **e–g**. Charge–charge or hydrogen-bonding and hydrophobic interactions are shown as gray dotted lines and spoked arcs, respectively. **h** Co-IP experiments testing the interactions between Ctr9 and Paf1-WT or mutants. The Paf1 (4S) mutant contains four amino acid substitutions L62S, M64S, V66S, and L68S. Extracts were prepared from HEK293T cells transfected with various combinations of plasmids, as indicated. Myc and GFP were tagged to the Paf1 and Ctr9, respectively. The bottom panel shows 3% of the Myc fusion proteins for each IP. Uncropped blots are shown in Supplementary Fig. 8

(folding) included mutations that were likely to affect the folding of each individual subunit. The third category (others) of mutations could not be well explained by the current structural model.

Given the extremely high conservation of these five amino acids located at the interface of the CTR9$^{(1-249)}$–PAF1$^{(57-116)}$ heterodimer from zebrafish to human (Fig. 2e and Supplementary Fig. 2), we next sought to test the effects of these disease category 1 (interface) mutations on the formation of the CTR9$^{(1-249)}$/PAF1$^{(57-116)}$ heterodimer. To this end, using the co-expression strategy, we noted that the purified mutant PAF1$^{(57-116)}$(I87M) or PAF1$^{(57-116)}$(E109K) did not form a stable complex with CTR9$^{(1-249)}$ (magenta lines in Supplementary Fig. 6b and SDS-PAGEs in Supplementary Fig. 6b1 and b2), nor did any purified CTR9$^{(1-249)}$ mutant (P164S, R201C, or A220T) form a stable complex with PAF1$^{(57-116)}$ (dashed cyan line or orange lines in Supplementary Fig. 6c and SDS-PAGEs in Supplementary Fig. 6c2-c4), in the analytical size-exclusion column. It was also noted that CTR9$^{(1-249)}$ WT or mutants are aggregated in size-exclusion column in the absence of PAF1$^{(57-116)}$ binding (Supplementary Fig. 6b and 6c). More, in the co-IP assays, it was clear that the I87M and E109K mutations of PAF1 and the P164S, R201C, and A220T mutations of CTR9 totally abolished the interaction between CTR9$^{(1-249)}$ and PAF1$^{(57-116)}$ (Fig. 3b). As a negative control, the category 3 (others) R98W mutation of CTR9 did not affect the interaction between CTR9$^{(1-249)}$ and PAF1$^{(57-116)}$, since the purified CTR9$^{(1-249)}$(R98W) mutant formed a stable complex with PAF1$^{(57-116)}$ (cyan line in Supplementary Fig. 6c and SDS-PAGE in Supplementary Fig. 6c1) and the Myc-CTR9$^{(1-249)}$(R98W) mutant specifically formed a complex with PAF1$^{(57-116)}$-GFP (lane 7 in Fig. 3b).

**Overall structure of the yeast Ctr9 and Paf1 heterodimer.** PAF1C is conserved throughout eukaryotes in its subunit compositions and functional importance[2,3,37], indicating that the structural features of each subunit and the assembly modes of holo- and/or sub-PAF1C are highly conserved among all eukaryotes. To evaluate this speculation, we investigated the structural features of the Ctr9/Paf1 heteromeric complex from *Saccharomyces cerevisiae* because PAF1C was initially identified in this species[1]. Accordingly, we analyzed the secondary structure predictions of yeast Ctr9 and Paf1. These predictions plus sequence alignment with the interaction regions of the human CTR9/PAF1 heterodimer (Fig. 2e and Supplementary Fig. 2), helped us identify the cognate interaction regions of the Ctr9/Paf1 complex through a 313-residue fragment of Ctr9 (aa 1-313, Ctr9$^{(1-313)}$), including TPR1-6, and a 70-residue fragment of Paf1 (aa 34–103, Paf1$^{(34-103)}$) (Fig. 4a). It was noted that Ctr9$^{(1-313)}$ and Paf1$^{(34-103)}$ assembled into a complex in the analytical size-exclusion column (red line in Supplementary Fig. 7a and SDS-PAGE in Supplementary Fig. 7a1). Next, we used the same approach by creating a single polypeptide through fusing Ctr9$^{(1-313)}$ to the N-terminus of Paf1$^{(34-103)}$ with a TEV-cleavable segment. The

purified single-chain fusion protein of Ctr9$^{(1-313)}$ and Paf1$^{(34-103)}$ (referred to as Ctr9$^{(1-313)}$-Paf1$^{(34-103)}$ hereafter) eluted as a single peak from an analytical size-exclusion column (black line in Supplementary Fig. 7a and SDS-PAGE in Supplementary Fig. 7a2), and subsequently, the crystals of Ctr9$^{(1-313)}$-Paf1$^{(34-103)}$ were successfully obtained.

The crystal structure of Ctr9$^{(1-313)}$-Paf1$^{(34-103)}$ was determined at a resolution of 2.53 Å with one molecule in the asymmetric unit (Supplementary Table 1). The entire artificial TEV-cleavable linker was also missing in the final model (Supplementary Fig. 3d and f), though SDS-PAGE analysis of dissolved native and Se-Met Ctr9$^{(1-313)}$-Paf1$^{(34-103)}$ crystals showed that these molecules were intact (Supplementary Fig. 3e). These results suggest that the artificial linker also did not alter the global structure of the Ctr9$^{(1-313)}$-Paf1$^{(34-103)}$. Notably, sequence alignment of Ctr9$^{(1-313)}$ and CTR9$^{(1-249)}$ indicated that the structure of yeast Ctr9$^{(1-313)}$ had two additional α-helices (α6B' and α7A') compared to human CTR9$^{(1-249)}$ (Supplementary Fig. 2), suggesting that these two α helices are not necessary for Ctr9 binding to Paf1. To test this hypothesis, a truncated fragment of Ctr9 (aa 1–270, Ctr9$^{(1-270)}$) was designed to test its binding to Paf1$^{(34-103)}$. As expected, size-exclusion chromatography and SDS-PAGE analysis showed that Ctr9$^{(1-270)}$ formed a heteromeric complex with Paf1$^{(34-103)}$ (blue line in Supplementary Fig. 7b and SDS-PAGE in Supplementary Fig. 7b1). Analytical ultracentrifugation further confirmed that the Ctr9$^{(1-270)}$/Paf1$^{(34-103)}$ complex formed a heterodimer with a stoichiometry of a 1:1 and with a molecular mass of ~36 kDa (Supplementary Fig. 7c).

Superimposition of the structures of the yeast Ctr9$^{(1-313)}$–Paf1$^{(34-103)}$ and human CTR9$^{(1-249)}$–PAF1$^{(57-116)}$ heterodimers resulted in a root-mean-square deviation of 2.63 Å for 198 equivalent Cα atoms (Supplementary Fig. 4c), indicating that human CTR9/PAF1 and yeast Ctr9/Paf1 subcomplexes share similar heterodimer structures. Notably, the sequence identity was low between the yeast Ctr9 or Paf1 and human CTR9 or PAF1 subunits (Fig. 2e and Supplementary Fig. 2); thus, the residues at each heterodimer interface were divergent (Figs. 2b–e and 4e–g). Similar to human CTR9/PAF1 subcomplex interactions, the Ctr9$^{(1-313)}$/Paf1$^{(34-103)}$ interface could be divided into three regions (Figs. 4e–g and 2e). In detail, region I was also mainly maintained by hydrophobic interaction through four conserved hydrophobic amino acids in Paf1 and corresponding amino acids in Ctr9, and by hydrogen bonds between D61/Q150 and K191, M64/K191, and D67/N104 (Fig. 4e and Supplementary Fig. 5d). Substitution of these four amino acids in Paf1 with serines (L62S, M64S, V66S, and L68S, referred to 4S) led to the complete disruption of Paf1/Ctr9 subcomplex formation (lane 3 in Fig. 4h). Region II of Paf1 was not conserved between yeast and other species (Fig. 2e). In this region, Paf1 had a longer loop than other species (e.g., worm to human), including a shorter α-helix (αC') that bound the N-terminal tail of Ctr9 (Fig. 4f and

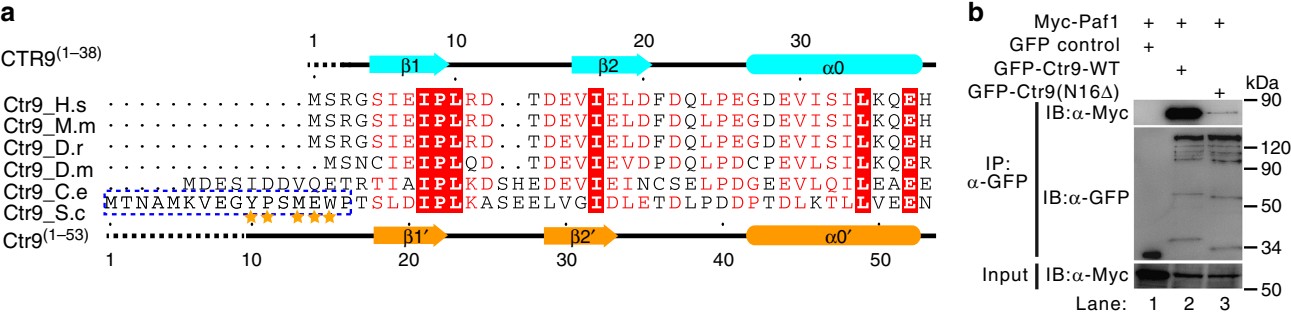

**Fig. 5** The longer N-terminal tail of Ctr9 is essential for its binding to Paf1. **a** Structural-based sequence alignment of the N-terminal fragments of Ctr9 in various species. In this alignment, the secondary structures of human CTR9$^{(1-38)}$ and yeast Ctr9$^{(1-53)}$ are shown at the top and bottom, respectively, and conserved residues are shaded in red. The amino acids 1–16 of yeast Ctr9, which were deleted in the GFP-Ctr9(N16Δ) construct [used in **b**], are marked with a dotted blue box. The amino acids Y10, P11, M13, E14, and W15 of Ctr9 involved in its binding to Paf1 are marked by orange stars. **b** Co-IP experiments testing the interactions between Ctr9 WT or Ctr9(N16Δ) mutant and Paf1. Extracts were prepared from HEK293T cells transfected with various combinations of plasmids, as indicated. The bottom panel shows 3% of the Myc fusion proteins for each IP. Uncropped blots are shown in Supplementary Fig. 8

Supplementary Fig. 5e), which is different from human CTR9/PAF1 complex. We noted that yeast Ctr9 had a longer N-terminal tail than other species (e.g., human CTR9) (Fig. 5a). In order to test the role of this longer N-terminal tail of Ctr9 in Ctr9/Paf1 complex formation, the most N-terminal 16 amino acids of Ctr9 were deleted in mutant [Ctr9(N16Δ)]. Notably, the [Ctr9(N16Δ)] mutant did not form complex with Paf1 thoroughly (Fig. 5b), indicating that this longer N-terminal tail of Ctr9 is essential to maintain the interaction between Ctr9 and Paf1. We also noted that residue L83 of Paf1 is only present in yeast (Fig. 2e), though L83 is involved in binding to Ctr9 (Fig. 4f). Substitution of L83 with serine (L83S) in Paf1 was not enough to disrupt the formation of the Ctr9/Paf1 subcomplex (lane 4 in Fig. 4h), indicating that residue L83 of Paf1 is not important for its interaction with Ctr9. For region III, D95 was the core amino acid in the C-terminus αD' of Paf1, which bound with R221, R243, and Y10 of Ctr9, mainly mediated by extensive hydrogen bonds (Fig. 4g and Supplementary Fig. 5f). D95K mutation in Paf1 also abolished the interaction between Ctr9 and Paf1 completely (lane 5 in Fig. 4h). Similar to human CTR9/PAF1 structure, it was also noted that the αA' of Paf1$^{(34-103)}$ is not involved in Ctr9$^{(1-313)}$/Paf1$^{(34-103)}$ heterodimer formation (Fig. 4b–d). Accordingly, a truncated fragment of Paf1 (aa 59–103, Paf1$^{(59-103)}$) was designed to delete the αA' of Paf1$^{(34-103)}$. Size-exclusion chromatography and SDS-PAGE analysis showed that Paf1$^{(59-103)}$ formed a complex with Ctr9$^{(1-313)}$ (cyan line in Supplementary Fig. 7b and SDS-PAGE in Supplementary Fig. 7b2), and a temperature-dependent denaturation assay demonstrated that the Ctr9$^{(1-313)}$/Paf1$^{(59-103)}$ complex is less stable than the Ctr9$^{(1-313)}$/Paf1$^{(34-103)}$ complex (cyan line comparing to red line and inset in Supplementary Fig. 7d), indicating that the αA' of Paf1$^{(34-103)}$ may contribute to the stability of the Ctr9$^{(1-313)}$/Paf1$^{(34-103)}$ complex. These structural analyses and mutagenesis-based data suggest that the specific recognition between Ctr9 and Paf1 results from a combination of numerous hydrophobic, charge–charge, and hydrogen-bonding interactions along the Ctr9 TPRs and the corresponding residues of Paf1.

**Ctr9 and Paf1 subcomplex is essential for Paf1C assembly.** Phenotypic analysis revealed that deletion either *Paf1* or *Ctr9* can influence yeast viability more severely than the other subunits of yeast Paf1C[54,55], and CTR9 is considered a key scaffold protein in assembling human PAF1C[46]. These findings, as well as our results demonstrating the conserved heterodimeric assembly of both human CTR9 and PAF1 and yeast Ctr9 and Paf1, along with the

finding that CTR9 and PAF1 are mutated in diseases, indicate that CTR9/PAF1 or Ctr9/Paf1 heterodimer formation may be important for complex assembly and function. To evaluate this hypothesis, we took yeast Paf1C as an example. To investigate the role of Ctr9/Paf1 subcomplex formation in the assembly of Paf1C, we used a co-IP strategy. In the WT Paf1C, both GFP-Ctr9 and GFP-Paf1 specifically co-immunoprecipitated with other Paf1C subunits (Fig. 6a, lane 2, and Fig. 6b, lane 2). In the mutant Paf1C, which contained mutants Myc-Paf1(4S) and Myc-Paf1(D95K), GFP-Ctr9 did not co-immunoprecipitate other Paf1C subunits (or amounts of some subunits to lesser degrees) (lanes 3 and 5 in Fig. 6a). In another experiment, significantly decreased Myc-Ctr9, 13×Myc-Rtf1, and Myc-Cdc73 and consistent amounts of Myc-Leo1 coprecipitated with both mutants GFP-Paf1(4S) and GFP-Paf1(D95K) (lanes 3 and 5 in Fig. 6b). Notably, the mutant Paf1(L83S) (which was capable of binding to Ctr9), formed complexes with other Paf1C subunits (lanes 4 in Fig. 6a, b). The asterisk in Fig. 6b indicate the degradation of Myc-Ctr9, according to the data shown in Supplementary Fig 7e. Collectively, these data indicate that Ctr9/Paf1 is the core component and this subcomplex formation is essential for assembling Paf1C. Together, these results, as well as the an architecture of Paf1C shown in prior study[45], led to the establishment of the interaction network for the assembly of yeast Paf1C (Fig. 6c).

**Ctr9 and Paf1 subcomplex is important for yeast viability.** The described structural and biochemical results thus far clearly demonstrate that the conserved Ctr9/Paf1 subcomplex formation is important for the assembly of yeast Paf1C. Next, we used yeast as a host to investigate the role of the Ctr9/Paf1 subcomplex in vivo. To this end, a *paf1Δ* strain was generated by replacing the *PAF1* gene with *S. cerevisiae URA3* (Supplementary Table 4). As expected, yeast cells with *paf1Δ* exhibited pleiotropic phenotypic traits, including slow growth, increased sensitivity to temperature, salt, and hydroxyurea (HU) (Fig. 7a–d), consistent with previous studies[54–56]. To further explore the role of the Ctr9/Paf1 subcomplex in vivo, we evaluated the importance of Ctr9/Paf1 subcomplex formation-mediated Paf1C assembly using a complementation assay. To evaluate their ability to complement *paf1Δ* phenotypes, the plasmids pP1K-PAF1(405), pP1K-PAF1(4S), pP1K-PAF1(L83S), and pP1K-PAF1(D95K) and the corresponding empty vectors were linearized and integrated into the *paf1Δ* strain. Interestingly, expression of both the WT *PAF1* (*PAF1*-WT) and the mutant *PAF1(L83S)* (which was capable of assembling Paf1C), but not the mutant *PAF1(4S)* or *PAF1(D95K)*

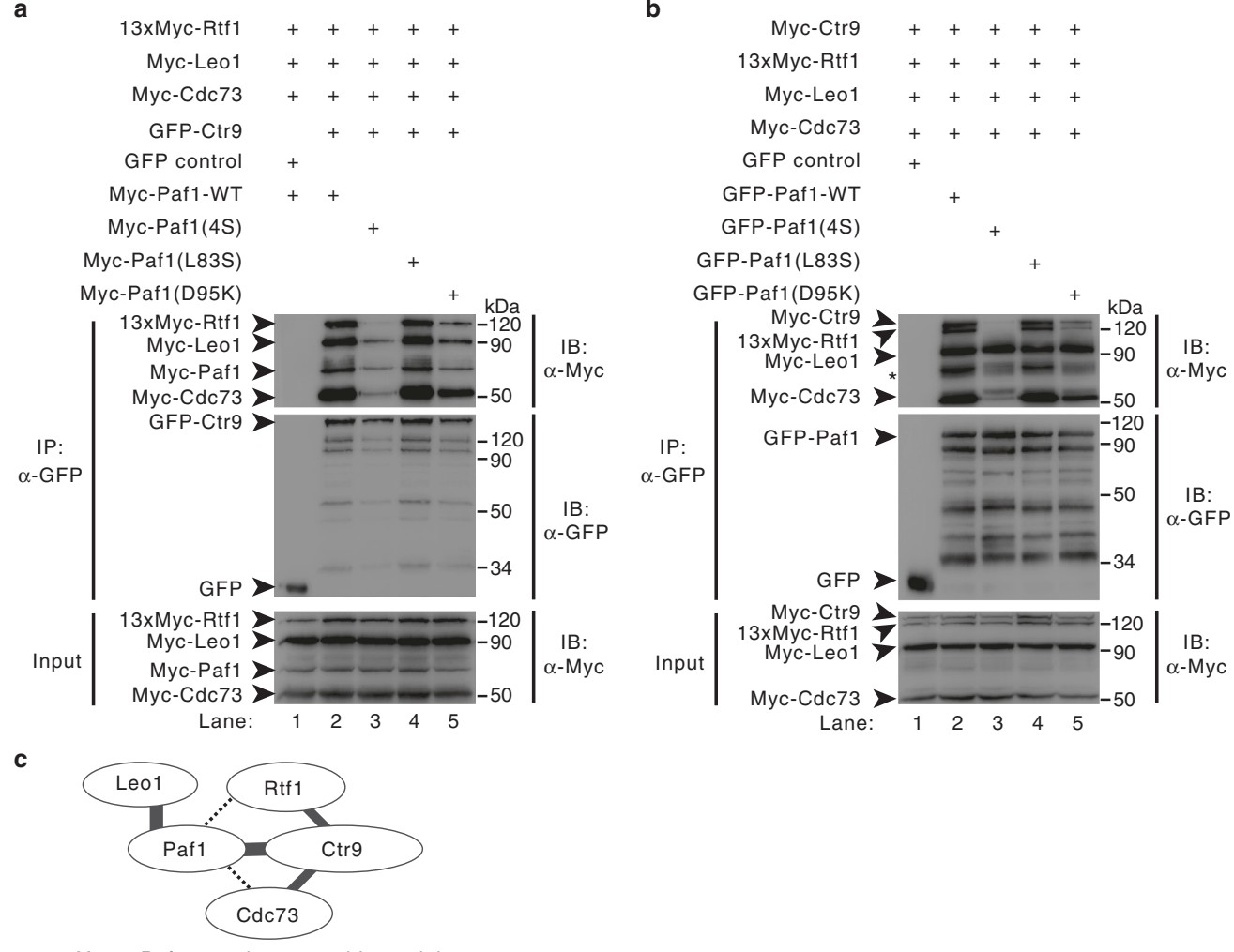

**Fig. 6** Ctr9/Paf1 subcomplex formation is essential for yeast Paf1C assembly. **a, b** Co-IP experiments of Paf1C formation by Ctr9 (**a**) and Paf1 (**b**). Extracts were prepared from HEK293T cells transfected with various combinations of plasmids, as indicated, immunoprecipitated with agarose-conjugated anti-GFP and subsequently immunoblotted with anti-Myc or anti-GFP, as indicated. The top panel shows the IP results. The middle panel represents the IP of GFP and GFP fusion proteins (GFP-Ctr9 or GFP-Paf1). The bottom panel shows 3% of the Myc fusion proteins for each IP. The asterisk indicate the degradation of Myc-Ctr9. **c** Model of yeast Paf1C assembly. The Ctr9/Paf1 heterodimer is the core component for Paf1C assembly. The bold line represents the interaction in the crystal structure of the Ctr9$^{(1-313)}$–Paf1$^{(34-103)}$ heterodimer or the interaction between Paf1 and Leo1 obtained from the IP results and studies[45,46]. The fine lines represent the interaction obtained from the IP results. The dotted lines represent interactions that need to be further studied. Uncropped blots are shown in Supplementary Fig. 8

(which were incapable of assembling Paf1C), fully rescued the *paf1Δ* phenotype (Fig. 7a–d). Collectively, these results indicate that Ctr9/Paf1 subcomplex formation-mediated Paf1C assembly is crucial for yeast growth.

Knockout of Paf1 (*paf1Δ*) in yeast cells leads to a strong, global decrease in synthesis of mRNA transcripts[45], which might be required for normal growth. Our results showed that Ctr9/Paf1 subcomplex formation-mediated Paf1C assembly was important for the growth of yeast (Fig. 7a–d). These findings, together with the fact that Paf1C is required to promote H3 methylation at K4, K36, and K79 in yeast[9–11], led us to consider whether the *paf1Δ* strain displayed pleiotropic phenotypes, including slow growth and increased sensitivity to temperature, salt, and HU because of a changed level of histone modification (e.g., H3K4 methylation). To test this hypothesis, we examined the level of histone methylation by analyzing the strength of the H3K4 methylation signal in whole yeast lysate. As expected, *paf1Δ* strains displayed drastically reduced levels of both the dimethylated and trimethylated H3K4 (referred to as H3K4me2 and H3K4me3, respectively), but not the monomethylated

H3K4 (H3K4me1), compared to the wild-type strain (lane 2 compared with lane 1 in Fig. 7e and the statistical data shown in Fig. 7f, g). These data are consistent with previous studies[9–11,57]. The expression of both PAF1-WT and the mutant PAF1(L83S) genes, but not the mutant gene PAF1(4S) or PAF1(D95K) (both were incapable of Paf1C assembly), restored both the H3K4me2 and H3K4me3 levels against a *paf1Δ* background (lanes 3 and 5 compared with lanes 4 and 6 in Fig. 7e and the statistical data shown in Fig. 7f, g). Together, these data indicate that Ctr9/Paf1 subcomplex formation is indispensable for Paf1C-mediated histone modification(s).

## Discussion

The crystal structures of human CTR9/PAF1 and yeast Ctr9/Paf1 subcomplexes presented in this study provide structural insights into the roles of Ctr9/Paf1 subcomplex formation-mediated Paf1C assembly and activity. To our knowledge, this is the paper to resolve crystal structures of the scaffold protein Ctr9 (albeit in part) at the atomic level. Though the two structures

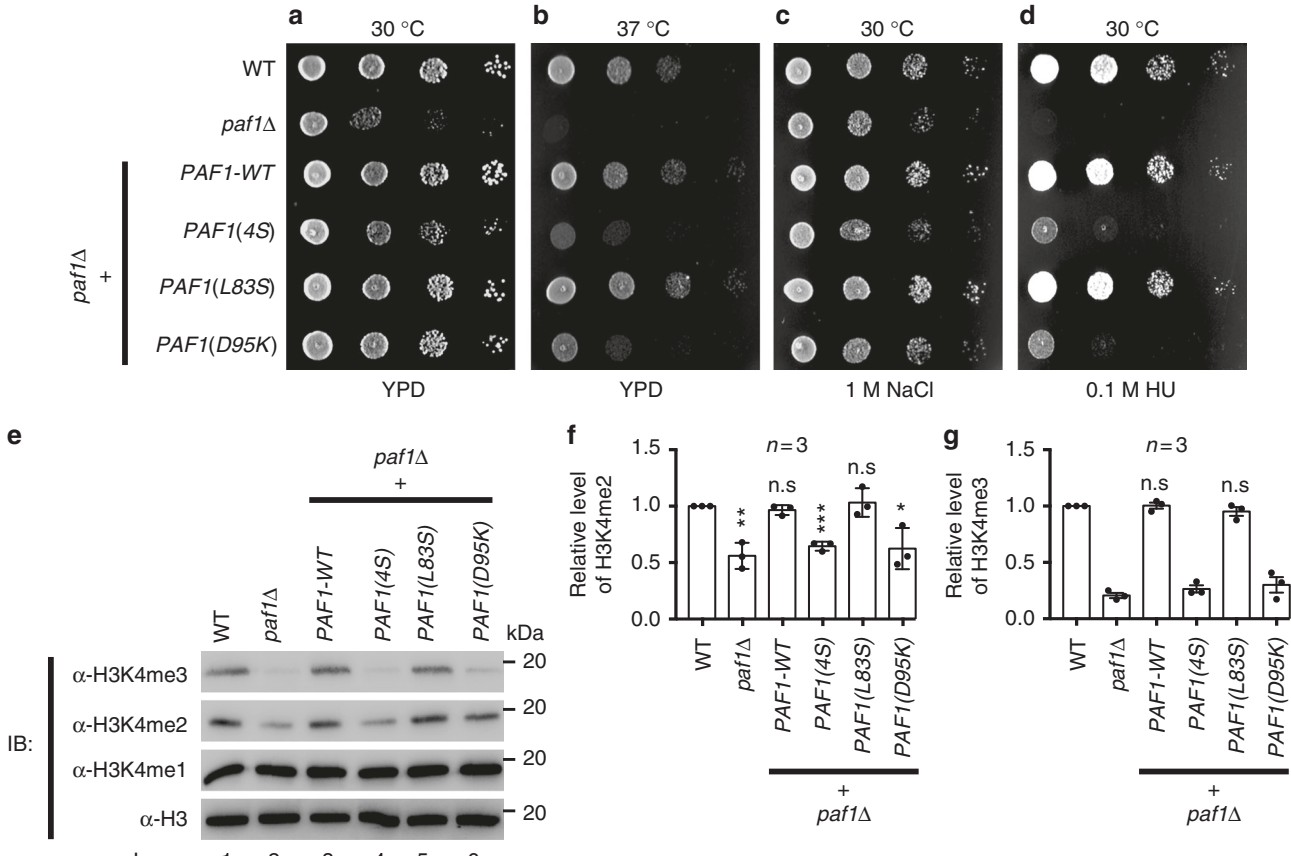

**Fig. 7** Ctr9/Paf1 subcomplex formation is essential for the integrity of a functional yeast Paf1C. **a–d** Ctr9/Paf1 subcomplex-mediated Paf1C assembly is essential for yeast cell viability. Strains of the indicated genotype were grown to late log phase/stationary phase (overnight) and plated in serial dilutions on a YPD plate. The plates are shown after 40 h of incubation at 30 °C (**a**), at 37 °C (**b**), on YPD containing 1 M NaCl (**c**), and on YPD containing 0.1 M hydroxyurea (HU) (**d**). **e** Representative immunoblot with specific H3K4me1/2/3 antibodies to detect methylation strength (top panels), or with H3 antibody to demonstrate equal loading (bottom panel). Bar graphs of H3K4me2 (**f**) and H3K4me3 levels (**g**) from various yeast strains, as indicated. All statistic data in this figure represents the results from three independent batches of experiments. Error bars represent s.e.m. ($n = 3$). $^{*}P < 0.05$, $^{**}P < 0.01$, $^{***}P < 0.001$. Uncropped blots are shown in Supplementary Fig. 8

determined in this study covered only 21% and 29% of the residues of full-length human CTR9 and yeast Ctr9, respectively, we believe that such structural information will help us to probe into the CTR9-incorporated PAF1C architecture. Our analysis revealed that both the hook-shaped (hookfold) PAF1[(57–116)] and Paf1[(34–103)] occupied the concave channel and convex surface of the right-handed superhelix-folded CTR9[(1–249)] and Ctr9[(1–313)], respectively (Figs. 1, 2, and 4), with very similar conformations (Supplementary Fig. 4c), although the sequence identity was low between human PAF1 and yeast Paf1 (Fig. 2e) or CTR9 and Ctr9 (Supplementary Fig. 2). Therefore, it can be assumed that the structural and biochemical features of Paf1/Ctr9 subcomplex described herein are shared by the orthologs of Paf1 and Ctr9 in other eukaryotic species.

Our previous study indicated that LEO1 binds to human PAF1C through PAF1 and that the CTR9 subunit is the key scaffold protein in assembling PAF1C[46]. In this study, we took yeast Paf1C as example to show that Leo1 binds to Paf1C also through Paf1 and further confirm that the Ctr9/Paf1 is the core component for Paf1C assembly and functional regulation (Figs. 6 and 7). Although these conserved structure and function, there exist some difference between human PAF1C and yeast Paf1C. For examples, human RTF1 is a less stable subunit of PAF1C[5,7,58], while the cognate yeast Rtf1 binds to Paf1C tightly

(lanes 2 in Fig. 6a, b); and the observation that the N-terminal tail of yeast Ctr9 (not human CTR9) is essential for its binding to Paf1 (Fig. 5). Further studies are necessary to reveal the evolutional structure and function or difference of this holocomplex.

TPR motifs are generally composed of α-helical elements and are involved in protein–protein interactions with great ligand-binding-mode diversity[47]. A typical example is that of an extended short-peptide binding conformation that can be observed in the Hop/Hsp90 interactions[59]. There, the ligand Hsp90 binds the concave side of the Hop TPR motifs. In another representative case, the ligand Caf4 forms a U-fold interacting with both the concave and convex sides of the Fis1 TPRs[60]. Notably, our structural analyses show that both ligands human PAF1 and yeast Paf1 formed a similar hookfold binding to both the concave channel (folded by the TPR1–3 and TPR4–5 subdomains of CTR9 or Ctr9) and the convex surface of TPR4–5 (Figs. 1, 2, 4, and Supplementary Fig. 5g), thus defining a, to the best of our knowledge, previously uncharacterized mode of interaction among TPR motifs.

Structural-based alignment indicated that the residues D79, G81, and D85 in Region I of human PAF1 are conserved from yeast to human or the residue D95 in Region II are conserved from worm to human (Fig. 2e), indicating these residues have important roles in complexes formation. It was noted that G81 (cognate residue G63

of yeast Paf1) induces a bend that is critical for L80 and V82 (cognate residues L62 and M64 of Paf1) to fit into their respective hydrophobic pockets, respectively (Supplementary Fig. 5a, d, h, i). Interestingly, the residues D79, D85, and D95 of PAF1 do not make contact with CTR9 (Fig. 2b, c), which is different to the cognate residues D61 and D67 involved in binding to Ctr9 (Supplementary Fig. 5h). However, these amino acids are very important for maintaining the local folding of PAF1. The side chain oxygen of D79 forms a hydrogen bond with the backbone nitrogen of G81; the side chain oxygen of D85 forms hydrogen bonds with the side chain hydroxyl oxygen of T91 and backbone nitrogen of I87 or N88; the side chain oxygen of D95 forms hydrogen bond with backbone nitrogen of N97 (Supplementary Fig. 5i). So we hypothesized that the disease-associated mutation D85N or D95Y (category 2 shown in Fig. 3a) will affect these hydrogen bonds and change the local folding of PAF1. To this end, we tested the mutants PAF1$^{(57–116)}$(D85N) and PAF1$^{(57–116)}$(D95Y) in their binding to CTR9. Size-exclusion chromatography and SDS-PAGE analysis showed the mutant PAF1$^{(57–116)}$(D85N) or PAF1$^{(57–116)}$(D95Y) can form a complex with CTR9$^{(1–249)}$ (Supplementary Fig. 1j and SDS-PAGEs in Supplementary Fig. 1j1 and j2). However, a temperature-dependent denaturation assay demonstrated that both the mutant complexes CTR9$^{(1–249)}$/PAF1$^{(57–116)}$(D85N) and CTR9$^{(1–249)}$/PAF1$^{(57–116)}$(D95Y) are less stable than CTR9$^{(1–249)}$/PAF1$^{(57–116)}$ WT complex (orange and magenta lines comparing to red line and inset in Supplementary Fig. 1i).

Size-exclusion chromatography and SDS-PAGE analysis showed that the purified human CTR9$^{(1–249)}$ is aggregated in absence of PAF1$^{(57–116)}$ binding (Supplementary Fig. 6). These data indicate that human CTR9/PAF1 subcomplex formation is essential for the folding of CTR9 (at least for CTR9$^{(1–249)}$).

PAF1C has been implicated in tumorigenesis[3]. In particular, PAF1 is involved in pancreatic and ovarian cancer[32,33], and CTR9 is involved in Wilms tumor and breast cancer[34,35]. These findings, together with the observation that Paf1/Ctr9 heterodimer formation is essential for Paf1C assembly and function (Figs. 6 and 7, and ref. 46), suggest that the disease-associated missense mutations (categories 1 and 2 shown in Fig. 3a) or other mutations in CTR9 (Supplementary Table 2) and PAF1 (Supplementary Table 3) may affect the integrity of a functional PAF1C, which affects gene regulation (e.g., via H3H4 di- and trimethylation) in the progress of these diseases. Future studies are necessary to evaluate this hypothesis.

## Methods

**Protein expression and purification**. To co-express a fragment of human CTR9 (residues 1–284, CTR9$^{(1–284)}$), full-length human PAF1 (residues 1–531, PAF1), and full-length human LEO1 (residues 1–666, LEO1), corresponding gene fragments were amplified by polymerase chain reaction (PCR) from the cDNA library of the human kidney epithelial (HEK) 293T cell line (CBTCCAS, GNHu17). PAF1 and CTR9$^{(1–284)}$ were cloned into two multiple cloning sites of pETDuet-1 vector (Novagen) separately and sequentially. LEO1 was cloned into an in-house modified version of the pET32a vector (Novagen). Ribosome binding sequence (RBS)-LEO1, which contains an RBS on the N-terminus of LEO1, was inserted sequentially before CTR9$^{(1–284)}$. All of the resulting proteins contained a thioredoxin (Trx)-his$_6$ tag on the N-terminus.

The tripartite complex recombinant proteins were expressed in BL21(DE3) Codon Plus *Escherichia coli* at 16 °C for 16–18 h. The cells were then lysed by high pressure cell cracker AH-1500 (ATS Engineering Limited). The Trx-His$_6$-tagged protein complex was purified by Ni-NTA affinity chromatography (QIAGEN) followed by size-exclusion chromatography on a HiLoad 26/60 Superdex 200 (GE Healthcare) in 50 mM Tris-HCl, pH 8.0, 200 mM NaCl, 1 mM EDTA, and 1 mM dithiothreitol (DTT). SDS-PAGE analysis of the recombinant Trx-CTR9$^{(1–284)}$/Trx-PAF1/Trx-LEO1 complex is shown in Supplementary Fig. 1a.

The similar co-expression strategies were used to prepare various truncations of both the human CTR9/PAF1 and yeast Ctr9/Paf1 complexes, including CTR9$^{(1–284)}$ and residues 57–266 of PAF1 (referred to as CTR9$^{(1–284)}$/PAF1$^{(57–266)}$, similar nomenclature hereafter), CTR9$^{(1–284)}$/PAF1$^{(57–140)}$, CTR9$^{(1–284)}$/PAF1$^{(57–116)}$, CTR9$^{(1–249)}$/PAF1$^{(57–116)}$, Ctr9$^{(1–313)}$/Paf1$^{(34–103)}$, and Ctr9$^{(1–270)}$/Paf1$^{(34–103)}$. All of the expression and purification processes were similar to the

procedures for the tripartite complex Trx-CTR9$^{(1–284)}$/Trx-PAF1/Trx-LEO1. The SDS-PAGE analyses of all recombinant protein complexes from human and yeast species are shown in Supplementary Fig. 1 and Supplementary Fig. 7, respectively.

To make a single-chain fusion protein of the CTR9$^{(1–249)}$/PAF1$^{(57–116)}$ and the Ctr9$^{(1–313)}$/Paf1$^{(34–103)}$ (hereafter named as CTR9$^{(1–249)}$-PAF1$^{(57–116)}$ and Ctr9$^{(1–313)}$-Paf1$^{(34–103)}$, respectively) complexes, DNA fragments were amplified by PCR and linked with a TEV protease-cleavable segment (Glu-Asn-Leu-Tyr-Phe-Gln-Ser). Two amino acids (Ser-Gly) were inserted both sides of the TEV protease-cleavable segment. The single open reading frame was cloned into an in-house modified version of the pET32a vector. The resulting protein contained a Trx-His$_6$ tag on its N-terminus.

The recombinant protein was expressed in BL21(DE3) Codon Plus *E. coli* cells at 16 °C for 16–18 h. The Trx-His$_6$-tagged protein was purified by Ni-NTA affinity chromatography, followed by size-exclusion chromatography on a HiLoad 26/60 Superdex 200 column in 50 mM Tris-HCl, pH8.0, 200 mM NaCl, 1 mM EDTA, and 1 mM DTT. After digestion with PreScission Protease to cleave the N-terminal Trx-His$_6$ tag, the target protein was purified on a Hiprep Q FF 16/10 anion-exchange column. The final purification step was size-exclusion chromatography on a HiLoad 26/60 Superdex 200 column in 20 mM N-2-Hydroxyethylpiperazine-N0-2-ethanesulfonic acid (HEPES), pH 7.5, 150 mM NaCl, 1 mM EDTA, and 1 mM DTT.

The Se-Met-substituted protein was expressed in methionine auxotrophic *E. coli* B834 (DE3) cells grown in LeMaster medium. All of the recombinant proteins were purified by Ni-NTA agarose affinity chromatography followed by ion-exchange and size-exclusion chromatography.

**Crystallization and data collection**. The crystals of human CTR9$^{(1–249)}$–PAF1$^{(57–116)}$ complex were grown at 4 °C at a protein concentration of 5 mg/mL using the sitting drop vapor diffusion method. The native and the Se-Met-substituted protein were all equilibrated against a reservoir solution of 0.1 M Tris-HCl, pH 8.0, 27.5% PEG 400 for 3 d. The crystals formation and growth was optimize using 27.5% 2-methyl-2,4-pentanediol as an additive. The crystals of the Se-Met-substituted yeast Ctr9$^{(1–313)}$–Paf1$^{(34–103)}$ complex were grown at 20 °C at a protein concentration of 30 mg/mL using the same method as above. The protein was equilibrated against a reservoir solution of 0.1 M Tris-HCl, pH8.5, 0.8 M Lithium sulfate, 0.01 M Nickel chloride and for 3 d. Then, we used the silver bullet additive (Hampton Research) containing 0.02 M HEPES-Na$^+$, pH 6.8, 0.004 M CaCl$_2$, 0.004 M MgCl$_2$, 0.004 M MnCl$_2$, and 0.004 M ZnCl$_2$ for further optimization to improve crystal quality. The crystals were frozen in a cryo-protectant that consisted of the reservoir solution supplemented with 30% mixture including 50% Ethylene glycol and 25% NDSB-201. All diffraction data were collected at the Shanghai Synchrotron Radiation Facility (SSRF) on beamlines BL17U[61] or BL19U using a CCD detector cooled to 100 K. All datasets were processed and scaled using the HKL2000 software package[62].

**Structure determination and refinement**. Phasing and initial model building of human CTR9$^{(1–249)}$–PAF1$^{(57–116)}$ complex crystal structure were determined by single wavelength anomalous dispersion (SAD) using PHENIX AutoSol wizard[63] and AutoBuild wizard[64], respectively. The initial phases and models of yeast Ctr9$^{(1–313)}$–Paf1$^{(34–103)}$ complex were determined by SAD using the Shelx C/D/E program[65]. Then, the initial models were further rebuilt and adjusted manually with Coot program[66] and were refined by phenix.refine program of PHENIX[67]. The final model was further validated using MolProbity[68]. The CTR9$^{(1–249)}$–PAF1$^{(57–116)}$ structure was refined with Ramachandran statistics of 96.1% favored, 3.9% allowed, and 0% outliers. The Ctr9$^{(1–313)}$–Paf1$^{(34–103)}$ structure was refined with 95.2% favored, 4.8% allowed, and 0% outliers. Detailed data collection and refinement statistics are summarized in supplementary Table 1. All structural figures were prepared using PyMOL (http://www.pymol.org/).

**Analytical ultracentrifugation**. SV experiments were performed in a Beckman Coulter XL-I analytical ultracentrifuge (Beckman Coulter) using double sector centerpieces and sapphire windows. An additional protein purification step on a HiLoad 26/60 Superdex 200 gel filtration column in 20 mM HEPES, pH 7.5, 150 mM NaCl, 1 mM EDTA, and 1 mM DTT was performed before the experiments. SV experiments were conducted at 42,000 rpm and 4 °C using interference detection. The SV data were analyzed using the SEDFIT program[69,70].

**Differential scanning fluorimetry**. Differential scanning fluorimetry (DSF) was performed on a CFX96 real-time PCR instrument (Bio-Rad). A 20 µL DSF solution is composed of 1–2 mg/ml of protein and 9× SYPRO Orange (Invitrogen) in the assay buffer containing 50 mM Tris, pH 7.5, 150 mM NaCl, 1 mM EDTA, and 1 mM DTT. During DSF assays, all samples were heated from 4 to 65 °C at a rate of 0.5 °C/min. Protein denaturation was monitored by the increased fluorescence signal of SYPRO Orange, which captures exposed hydrophobic residues during thermal unfolding. Fluorescence was measured using the VIC channel of the CFX96. The recorded melting curves were analyzed by the CFX Manager software (Bio-Rad). The temperature corresponding to the point of inflection was defined as the melting temperature (Tm).

**Cell culture and transfection**. HEK293T (CBTCCAS, GNHu17) cells were cultured in DMEM (Sigma, USA) supplemented with 10% (vol/vol) FBS (Biological Industries) at 5% $CO_2$ and 37 °C. HEK293T cells were transfected with Polyethylenimine (Polysciences, Inc.) according to the manufacturer's protocol.

**Co-immunoprecipitation**. HEK293T cells were transfected with the indicated combinations of plasmids. After 24 h transfection, HEK293T cells were lysed using ice-cold cell lysis buffer (50 mM Tris-HCl, pH 7.4, 150 mM NaCl, 3% glycerol, 0.5% NP40, 0.5% Triton X-100, 1 mM phenylmethylsulfonyl fluoride, and protease inhibitor cocktails) and cleared by centrifugation at 13,000 rpm for 20 min at 4 °C. The supernatants were then incubated with agarose-conjugated anti-GFP antibody for 30 min at 4 °C. The agarose beads were washed three times with cell lysis buffer and eluted with SDS sample buffer. Samples were then subjected to SDS-PAGE and western blot analysis.

**Western blotting**. The proteins were separated by SDS-PAGE and transferred to polyvinylidene difluoride (PVDF) membrane (Millipore). The membranes were subsequently blocked with 10% nonfat milk in TBST (50 mM Tris-HCl, pH 7.4, 150 mM NaCl, and 0.1% Tween 20) for 1 h. The PVDF membranes were immunoblotted with anti-Myc antibody (Sigma, M4439) diluted 1:5000 and anti-GFP (Sigma, G1544) antibody diluted 1:5000 at room temperature for 1 h, and then probed with horseradish-peroxidase-conjugated secondary antibodies with a dilution of 1:5000 (Santa Cruz, sc-2004 and sc-2005) and developed with a chemiluminescent substrate (Millipore). Protein bands were visualized on the Tanon-5200 Chemiluminescent Imaging System (Tanon Science and Technology). All experiments were repeated at least three times.

**Yeast culture and growth conditions**. Yeast cells were grown in rich (YPD; 1% yeast extract, 2% peptone, and 2% glucose) or synthetic minimal medium (synthetic minimal media; 0.67% yeast nitrogen base, 2% glucose, and amino acids as needed) at 30 or 37 °C. In some experiments, cells were also cultured in the presence of 1 M NaCl or 100 mM HU.

**Yeast strains**. The *S. cerevisiae* strains used in this study are summarized in Supplementary Table 4. For gene disruption, the *PAF1* coding regions were replaced with the *S. cerevisiae URA3* using PCR primers containing 60 bases of identity to the regions flanking the open reading frame (ORF). The forward primer sequence is 5'-GTACAATAGAACAGTGCTCATAATAGTATAAAGGGTCA-CAGCTTCGTACGCTGCAGGTCG-3', and the reverse primer sequence is 5'-CAGGTTTAAAATCAATCTCCCTTCACTTCTCAATATTCTAGCA-TAGGCCACTAGTGGATC-3'. To construct a plasmid expressing WT *PAF1* under the control of the endogenous promoter, a DNA fragment containing the *PAF1* locus (including 1000 base pairs of the *PAF1* 5' sequence and the *PAF1* ORF) was amplified from *S. cerevisiae* genomic DNA. The corresponding variants Paf1 (4S), Paf1(L83S), and Paf1(D95K) were constructed using site-directed mutagenesis. All of the recombinant fragments were inserted into vectors pRS405 using BamHI and XhoI sites. The resulting plasmids pP1K-Paf1(405), pP1K-Paf1(4S) (405), pP1K-Paf1(L83S)(405), and pP1K-Paf1(D95K)(405) were linearized by SnabI before transforming into *paf1Δ* strain. When necessary, empty integration plasmids were transformed into appropriate yeast strains to ensure that strains being compared possessed the same auxotrophic genotype.

**Histone methylation in vivo**. Yeast cells were grown at 30 °C in YPD medium to stable log phase ($OD_{600} \approx 0.6$–1.0). Then, cells were harvested and washed in 1 ml zymolyase buffer (50 mM Tris-HCl at pH 7.5, 10 mM $MgCl_2$, and 1 M sorbitol) once. The yeast cells were resuspended with 470 μl zymolyase buffer, 5 μl DTT and 25 μl zymolyase and incubated at 30 °C for 30 min. Then, pellets were collected and resuspended with 600 μl EBX (20 mM Tris-HCl at pH 7.4, 100 mM NaCl, 0.25% NP40, 0.5% Triton X-100, 50 mM Na-butyrate) on ice for 15 min. The lysates were centrifuged, and the resulting pellets were resuspended with 900 μl NIB (20 mM Tris-HCl at pH 7.5, 1.2 M Sucrose, 100 mM NaCl) on ice for 15 min. After discarding the supernatant, EBX containing 1% Triton X-100 was added into the pellets for another 15 min on ice. Then, the chromatin and nuclear debris were obtained. These were washed with EBX four times and added to loading buffer for SDS-PAGE, followed by immunoblotting with anti-H3 (Abcam, ab1791), H3K4me1 (Abcam, ab8895), H3K4me2 (Abcam, ab32356) and H3K4me3 (Abcam, ab8580) antibodies with dilutions of 1:2000, 1:500, 1:2000, and 1:1000, respectively.

## Data availability

The authors declare that all data supporting the findings of this study are available within the article and its supplementary information files or form the corresponding author upon reasonable request. Coordinates of the crystal structures of human CTR9/PAF1 and yeast Ctr9/Paf1 have been deposited in the Protein Data Bank under the accession code 5ZYQ and 5ZYP, respectively.

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

## Acknowledgments

We are grateful to the staff at the beamlines BL17U1 and BL19U1 of the Shanghai Synchrotron Radiation Facility (SSRF) for excellent technical assistance during data collection. We thank Dr. Jianchao Li (HKUST, Hong Kong) for his help in structure determination. This work was supported by the 973 Program 2014CB910201 (to J.L.), National Natural Science Foundation of China [31470755 and 31670758 (to J.L.); 31870750 to (to H. Z.)].

## Author contributions

Y.X., M.Z., H.Z., and J.L. designed research. Y.X., M.Z., X.C., Y.C., H.X., J.W., and H.Z. performed research. Y.X., M.Z., H.Z., and J.L. analyzed the data and prepared the figures. Y.X., M.Z., H.Z., and J.L. wrote the paper. All authors reviewed the manuscript. J.L. coordinate the research.

## Additional information

**Competing interests:** The authors declare no competing interests.

