## [Peer Review File · Nature Communications]

Reviewers' comments:

Reviewer #1 (Remarks to the Author):

The authors report high-quality crystallographic structures of the N-terminal region of Ctr9 bound to a N-terminal tether in the Paf1 subunit. Our structural understanding of the Paf1 complex is still very fragmented and therefore these structures, for the yeast and human complexes, are important. The authors add some functional data to demonstrate the relevance of the observed interaction. This reviewer supports rapid publication of these competitive structures in Nature Communications after the following minor concerns have been addressed.

-It is unclear how the authors built their structures. Lines 471-472: do the authors use the models generated by AutoSol/Autobuild as initial models in Coot or do they build the structure de novo into the electron density? This should be clearly described.

-It is not clear whether CTR9 1-249 is folded/well behaved on its own. If not, this would explain the size exclusion results presented in Fig S6. Lack of interaction with PAF1 results in poor solubility/behaviour of CTR9. It appears that most PAF1 mutants made with CTR9 elute in the void volume of the column. If this is true, the authors should state this in the text. Labels should not lie above the gel images.

-Human protein names should be written in all caps ie PAF1, LEO1, etc. Yeast names are written as Leo1, Paf1, etc. This will simplify the text and follows established naming conventions.

-Authors used MS and limited proteolysis to identify interacting regions of PAF1 and CTR9 (lines 97-98). Both procedures should be described in the methods section.

-Figure 3- The legend should state what the Ctr9 N16 Δ and Paf1 (4S) mutants are (ie amino acids mutated).

-Figure 4A: Leo1 appears to be weakly associated with all complexes the authors immunoprecipitate. The Leo1 levels in the total lysate appear normal. Is there an explanation for this? The figure legend for the bottom panel should state that the blot is representative of 3% of the input used for the IP, or something to this effect. Please add a short statement on your interpretation of this.

-Figure 4B: There is an additional band in the Myc blot, under LEO1. Do the authors know the identity of the band? This is not essential but maybe interesting.

-Figure S1: The nomenclature for heterodimers and fusion proteins is confusing.

-Line 96: change phrase to "eluted from size exclusion as a tripartite complex" or something similar

-line 141 "TRP" should be "TPR"

-line 238 "Se-substituted" should be defined. It appears that the authors used Se-Met for phase determination, and this language should also be used in the text.

-line 411: the authors should state what the AH-1500 does. Is it a sonicator, French press, etc.

-Line 430: It is unclear if the Ser-Gly additions are found at the N- or C- terminal side of the TEV site

-Line 457: state as "the silver bullet additive" instead of "the silver bullet" and state manufacturer

-Line 494: °C appears to have been replaced by a box in a number of positions.

Reviewer #2 (Remarks to the Author):

This manuscript describes a structure-function analysis of the association between the Paf1 and Ctr9 subunits of the Paf1 complex (Paf1C). Paf1C plays an important, albeit not fully understood, role in transcription elongation and connecting elongation to chromatin modifications, particularly activating marks on histone tails. Paf1C is critical for viability in yeast and has important connections with human disease. For example, some of the subunits and their interactions are thought to have tumor suppressor activity. There are few structural details about how the complex assembles and how the complex interacts with transcriptional machinery.

The authors have solved novel high-resolution crystal structures of two similar complexes, from human and yeast, containing the primary interacting regions of Paf1 and Ctr9. They compliment the structure with biochemical interaction studies in human cells and functional studies in yeast that exploit structure-based mutations. The main significant findings are the structural details of the Paf1-Ctr9 association, the mapping of several cancer-derived mutations to the interface, and the functional studies that demonstrate the Paf1-Ctr9 interface is critical for Paf1C assembly and function. The technical quality of the crystal structures is high and conclusions are properly supported by the complimentary biochemical data. The study is for the most part rigorously done and well explained. Considering the importance of the complex and lack of structural details currently available, the work will have impact. It should be published after the authors are given the opportunity to consider the following minor concerns and suggestions:

1) It would be useful if the authors could explain more how they envision this subcomplex fitting into the full Paf1C complex. Although the previously published low resolution EM map may not allow for unambiguous docking of the high resolution structure, the authors could at least comment on different relationships between the subunits within the complex and perhaps show a schematic model. For example, the Ctr9 binding site in Paf1 is near the Leo1 binding site. Does Leo1 also contact Ctr9? More broadly, how is the observation that this local interaction is required for overall complex assembly consistent with a model for architecture?

2) Line 84: The Ctr9-Caf1 heterodimer is called an "obligate" heterodimer, but it is not made clear what evidence supports this description. Was this published elsewhere? Fig S6 suggests that Ctr9 is aggregated in the absence of Paf1 association, but this part of the observation is not given any significant attention.

3) Lines 105-106: 1:1 stoichiometry cannot be rigorously determined from size-exclusion column (SE), particularly without standards. However, the 1:1 stoichiometry is convincingly shown by the molecular weight measurement using analytical ultracentrifugation data (and crystal structure). It would be more justified to change the conclusion from the SEC data to focus on the co-elution and the fact that the heterodimer and fusion elute in the same place, which supports a similar folded structure.

4) Line 141: Change "five TRP" to "five TPR".

5) It is curious that the A helix in Paf1 does not make contact with Ctr9 or anything that is clear in Fig. 2, yet it is ordered and somehow stabilized as a helix. Furthermore, there are some highly

conserved residues and a disease related mutation in this helix. Does this helix contact a different Ctr9 in the crystal lattice that is related by symmetry? If so, this begs the questions as to whether it could also make the equivalent interaction in solution. Does deletion of the A helix not change heterodimer stability? Do the authors have other thoughts on the role of this helix?

6) The authors could comment on the role of some of the most conserved residues in Paf1 Region I that do not appear to make contact with Ctr9 (D79, G81, and D85). For example, it appears in the view in Fig. S5A that G81 induces a bend that is critical for both L80 and V82 to fit into their respective hydrophobic pockets.

7) If possible, it would be a more impactful experiment if the co-IP in HEK cells were performed with full-length proteins. It is not surprising that the structure-derived mutations show the predicted effects when using identical constructs to those in the crystals. The more important question is whether the structural results are applicable in the context of the full-length proteins.

Reviewer #3 (Remarks to the Author):

In the manuscript entitled "Paf1/Ctr9 subcomplex formation is essential for Paf1 complex assembly and functional regulation" Xie et al. present the structure of a subcomplex of the human and yeast Ctr9/Paf1. This subcomplex is part of the polymerase-associated factor 1 complex (PAF1C), which is involved in transcriptional elongation and chromatin regulation. There is limited structural information about PAF1C. This work builds up on previous work (Chu et al., 2013; Xu et al., 2017) and constitutes a step forward in our understanding of the complex formation at the molecular level. Furthermore, by using a combination of biophysical and biochemistry methods, the authors demonstrate that Ctr9/Paf1 is required for efficient assembly of PAF1C. In addition to the new structural data, the authors perform key mutations on the interface of Ctr9/Paf1 and monitor complex assembly and functionality in vivo assays. These point mutations confirm the impact that the correct assembly of PAF1C might have on cancer progression, since some of the previously identified mutations in cancer disease play also a key role in complex formation.

Overall, the science, methods and statistics presented in this work appear sound. There are no major issues, however some further clarification is needed on the points given below. Additionally, the paper would be easier to understand if the presentation was improved, e.g. simplification of figures.

1. Quality of Figures 4A, 4B and 5E is poor. Replacement with higher resolution images is a necessity.

2. The authors perform co-immunoprecipitations (co-IP) experiments to test the interaction between hCtr9 and hPaf157-116. In Figure 2F, these assays are shown. In light of the crystal structure (hCtr91-249 - hPaf157-116), an extra co-IP assay showing the result of the interaction between hCtr91-249 and hPaf157-116 would complete Figure 2F (similar to lane 2 in Figure 2H). Alternatively, the authors should mention Figures 2F and 2H together whenever they refer to this experiment.

3. The authors attempt to define a new mode of interaction among TPR motifs. TPR motifs are widespread in nature and present a common fold. Further evidence would be needed to conclude that a novel TPR-SLIM (small linear motif) interaction has been found. This point seems over-interpreted. Comprehensive bioinformatics analysis and a figure showing a comparison of structural results (and different type of folds) would be necessary to raise this conclusion. Otherwise, the authors are encouraged to moderate (or even omit) this statement in the text.

4. To validate the interactions observed in the structure of hCtr91-249/hPaf157-116 a number of mutagenesis studies are presented. The overall conclusion of this study seems precise. However, Figure S6 and related comments need to be improved for clarity. For example, all mutants (except hCtr91-249 (R98W)/hPaf157-116, Figure S6C, blue lane) seem to aggregate. This is a typical

behaviour of disrupted complexes that occurs when one or more of the interactor partners are missing. In Figures S6A and S6A2, Ctr9 seems to appear in the void volume (first peak of SEC, S6A) and Paf1 shows up later (second peak of SEC, S6A). In the same figure, it also seems like Ctr9 is also present in this second peak (Figure S6A2, lane 35) and shows the same intensity as Paf1. This does not correlate with the absorbance of the chromatogram peaks. In addition, in S6A2 there are 2 bands of similar size (14.4 kDa) and it is unclear which one corresponds to hPaf157-116(5M). Could the authors elaborate further on the interpretation of the chromatograms and corresponding gels, and specifically on Figures S6A and S6A2? Could it be that mutated complexes are still observed in the second peak of the chromatograms (~fraction 35) but with disrupted stoichiometry? See comments on how to improve Figure S6A below.

5. Lines 85-86 "elucidated the mechanism responsible for human diseases caused by mutations of Paf1 and Ctr9" are an overstatement of the conclusions. The authors have demonstrated that the assembly of PAF1C might have an impact on specific cancer diseases, but further investigation is needed to elucidate the mechanism responsible for the human disease.

Suggestions to improve figures.

Figure 1A.

- Summarizing disease-associated mutations on this figure is not constructive. A table showing a summary of mutations is recommended. For simplicity, information about mutations should be omitted in the figure. This way, this figure would compare easily to Figure 3A. Instead, the schematic representation should show all domains present in hCtr9 and hPaf1 (full domain representation) for clarity. Schematic representation of Figure 3A should be modified accordingly.

Figures 1B-D.

- Labels for hCtr91-249 and hPaf157-116 should be added to the cartoon representations for easy interpretation of the figure.
- It would be easier to interpret the structural panels if they all show the protein structure in a similar manner, eg. cylinders vs alpha helices. This would make it easier to see how the various views are related to each other.
- The same applies to Figures 3B-C.

Figure 2.

- Figures 2B, 2C and 2D. Interpretation of these figures is difficult. A LigPlot or a cartoon representation with a zoom of the highlighted areas showing the residues involved in the interactions would clarify the figure considerably. The same applies to Figures 3D-F.
- Figure 2E. The name of the proteins used to depict secondary structures on top (hPaf1) and bottom (yPaf1) of alignment should be indicated for clarity. Have these secondary structures been predicted with any specific program? The same applies to Figure S2 (hCtr9 and yCtr9).
- Figures 2F and 2H. Labels of specific band proteins are not shown in Figures 2F and 2H. Also, is hPaf157-116 tagged with GFP in N or C terminus? (N-terminal residue in Figure 2F and C-terminal one in Figure 2H).
- Figure 2G. To make it easier understand what type or residues are included in categories 1, 2, 3 (both mentioned in the text and in this figure), these could be renamed as 'interface', 'folding' and 'others', respectively. This figure is difficult to interpret and it is not very informative (again a comprehensive table with mutations would help if added). To make this figure more useful, the authors could focus on showing just the mutations used in this work (categories 1 and 3).

Figure 5E. Label of IB is missing.

Figure S1.

- Contrast of S1B needs to be improved.
- Color code for chromatograms is not shown in S1E.

Figure S3.

This figure can be used to highlight the differences in the N-terminal domain of the human and yeast proteins, which will complement Figure 3H. This represents an important finding in this paper, i.e. this N-terminal region is essential to maintain the interaction between yCtr9 and yPaf1.

Figure S4C.

To appreciate the differences between human and yeast Ctr9/Paf1, the use of only two colours would simplify the representation.

Figure S5.

- S5A and S5B. It is not possible to appreciate the stick representation of residues very well. A zoom highlighting these regions would help the reader. Labels in green and blue are not easy to read. A possibility could be to use lighter colours.
- Figure S5E. Not mention in the text.

Figure S6.

- Figures S6A, B, C. Label for "fractions" in chromatograms as well as asterisks on top of those fractions shown in correspondent gels are necessary to clarify figure.
- Gels. Represent fraction number in just one lane would help reading of the figure.

Minor points.

- Several parts of the manuscript need to be clarified so it is more suitable for a general audience. This will aid the reader understand better the text. For example, a general overview of what it is known about Paf1, Ctr9 and PAF1C from the structural point of view is missing.
- 'Crystal structure of the hCtr9/hPaf1 heterodimer' section. It is unclear how the TEV constructs have been made. Is the TEV site cleaved at all? Specify in line 117 where the TEV-cleavable segment has been included. Explain how the artificial linker is helping crystallization.
- Lane 267. Elaborate further on the interaction between the N-terminal tail of yCtr9 and yPaf1 and how this is different to the human system.
- Lane 268. Specify that yL83 is tested because it is only present in yeast.
- Different terms are used for co-immunoprecipitation (co-IP). For example in lines 290 and 294 it is named as coprecipitation. Check consistency for this term.
- A cartoon showing the updated proposed assembly of PAF1C is recommended. This would enhance the result and discussion.
- D95K mutation in yPaf1 abolished the interaction between yCtr9 and yPaf1 completely. It would be interesting to test if D104K in hPaf1 had the same effect.

Point-by-point responses to the reviewers' comments:

(Our responses and all changes in the revised manuscript are in blue)

Reviewer #1 (Remarks to the Author):

The authors report high-quality crystallographic structures of the N-terminal region of Ctr9 bound to a N-terminal tether in the Paf1 subunit. Our structural understanding of the Paf1 complex is still very fragmented and therefore these structures, for the yeast and human complexes, are important. The authors add some functional data to demonstrate the relevance of the observed interaction. This reviewer supports rapid publication of these competitive structures in Nature Communications after the following minor concerns have been addressed.

We thank the reviewer for supporting to publish our work in Nature Communications, and for the constructive critiques and suggestions below.

-It is unclear how the authors built their structures. Lines 471-472: do the authors use the models generated by AutoSol/Autobuild as initial models in Coot or do they build the structure de novo into the electron density? This should be clearly described.

Thanks for pointing out this unclear description. Phasing and initial model building of human CTR9⁽¹⁻²⁴⁹⁾-PAF1⁽⁵⁷⁻¹¹⁶⁾ complex crystal structure were determined by single wavelength anomalous dispersion (SAD) using PHENIX AutoSol wizard and AutoBuild wizard, respectively. The initial phases and models of yeast Ctr9⁽¹⁻³¹³⁾-Paf1⁽³⁴⁻¹⁰³⁾ complex were determined by SAD using the Shelx C/D/E program. Then, the initial models were further rebuilt and adjusted manually with Coot program and were refined by phenix.refine program of PHENIX. The final model was further validated using MolProbity. We have modified the method accordingly.

-It is not clear whether CTR9 1-249 is folded/well behaved on its own. If not, this would explain the size exclusion results presented in Fig S6. Lack of interaction with PAF1 results in poor solubility/behaviour of CTR9. It appears that most PAF1 mutants made with CTR9 elute in the void volume of the column. If this is true, the authors should state this in the text. Labels should not lie above the gel images.

Agree with the reviewer's comments. The purified human CTR9⁽¹⁻²⁴⁹⁾ recombinant protein was aggregated in size-exclusion column (blue line in **Fig. I**, below for convenience), comparing to CTR9⁽¹⁻²⁴⁹⁾/PAF1⁽⁵⁷⁻¹¹⁶⁾ complex protein (black line in **Fig. I**). Additionally, we found that yeast Ctr9⁽¹⁻³¹³⁾ forms inclusion bodies when expressing in *E. coli* (data now shown). These data indicate that both human CTR9/PAF1 and yeast Ctr9/Paf1 subcomplexes formation are essential for the folding of CTR9 and Ctr9 (at least for CTR9⁽¹⁻²⁴⁹⁾ and Ctr9⁽¹⁻³¹³⁾), respectively. We have included **Fig. I** as the revised **Supplementary Fig. 6a** and **a1-a3** and modified the manuscript accordingly.

Fig. 1. CTR9⁽¹⁻²⁴⁹⁾ is aggregated in the absence of PAF1⁽⁵⁷⁻¹¹⁶⁾ binding. Analytical gel filtration profiles and SDS-PAGE of CTR9⁽¹⁻²⁴⁹⁾/PAF1⁽⁵⁷⁻¹¹⁶⁾ (black line in a and a1), CTR9⁽¹⁻²⁴⁹⁾/PAF1⁽⁵⁷⁻¹¹⁶⁾(5M) (dotted black line in a and a2), and CTR9⁽¹⁻²⁴⁹⁾ (blue line in a and a3). Those fractions shown in correspondent gels are indicated by a two-way arrow.

-Human protein names should be written in all caps ie PAF1, LEO1, etc. Yeast names are written as Leo1, Paf1, etc. This will simplify the text and follows established naming conventions.

Thanks for suggestion. Following the reviewer's suggestion, we have modified the manuscript accordingly. Additionally, complex names of human and yeast species are named as PAF1C and Paf1C, respectively.

-Authors used MS and limited proteolysis to identify interacting regions of PAF1 and CTR9 (lines 97-98). Both procedures should be described in the methods section.

Thanks for suggestion. Actually, after finishing limited proteolysis using trypsin, we found that the Mass spectrometer was off service. At that time, according to the results of limited proteolysis, together with our previous results shown in Ref. #46 in this manuscript and the sequence alignment of PAF1 from various species, we tested the region in PAF1 (aa 57-266) whether can bind to CTR9⁽¹⁻²⁸⁴⁾ or not independent of the LEO1 subunit. Fortunately, we got the positive result. We have modified the manuscript accordingly.

-Figure 3- The legend should state what the Ctr9 N16Δ and Paf1 (4S) mutants are (ie amino acids mutated).

Following the reviewer's suggestion, we have modified the figure legends in which state amino acids substitutions or truncation in mutants PAF1⁽⁵⁷⁻¹¹⁶⁾(4S), PAF1⁽⁵⁷⁻¹¹⁶⁾(5M), and Paf1(4S) or Ctr9(N16Δ), in the revised **Fig. 2f**, **Fig. 3h** or **Fig. 4**, respectively.

-Figure 4A: Leo1 appears to be weakly associated with all complexes the authors immunoprecipitate. The Leo1 levels in the total lysate appear normal. Is there an explanation

for this? The figure legend for the bottom panel should state that the blot is representative of 3% of the input used for the IP, or something to this effect. Please add a short statement on your interpretation of this.

We agree with the reviewer on this comment and thank the reviewer for pointing this issue out. To figure this query out, we have optimized the co-IP experiments and repeated at least three times (representative data in **Fig. II**, below). The newly acquired data indicate that Leo1 binds to Paf1C through Paf1 independent of Ctr9/Paf1 subcomplex formation (lane 3 comparing with lane 2 in **Fig. IIb**). The asterisk in **Fig. IIb** indicate the degradation of Myc-Ctr9 (detailed data shown in **Fig. III**, below). We have included **Fig. II** as the revised **Fig. 5** in the revised manuscript and modified the manuscript accordingly.

Figure II. Ctr9/Paf1 subcomplex formation is essential for yeast Paf1C assembly.

(a and b) Co-IP experiments of Paf1C formation by Ctr9 (a) and Paf1 (b). Extracts were prepared from HEK293T cells transfected with various combinations of plasmids, as indicated, immunoprecipitated with agarose-conjugated anti-GFP and subsequently immunoblotted with anti-Myc or anti-GFP, as indicated. The top panel shows the IP results. The middle panel represents the IP of GFP and GFP fusion proteins (GFP-Ctr9 or GFP-Paf1 or their mutants). The bottom panel shows 3% input of the Myc fusion proteins used for each IP. The asterisk indicate the degradation of Myc-Ctr9.

-Figure 4B: There is an additional band in the Myc blot, under LEO1. Do the authors know the identity of the band? This is not essential but maybe interesting.

Thanks for pointing this issue out. To answer this question, we have examined the identity of the additional band via a co-IP strategy and confirm that this band is the degradation product of Ctr9, since this band will appear in the context of co-IP assays adding Ctr9 (lane 3 comparing with lane 2 in **Fig. III**, below). We have included **Fig. III** as the revised Supplementary **Fig. 7e** and modified the manuscript accordingly.

Fig. III. Co-IP experiments by GFP-tagged yeast Paf1 (GFP-Paf1). Extracts were prepared from HEK293T cells transfected with various combinations of plasmids, as indicated, immunoprecipitated with agarose-conjugated anti-GFP and subsequently immunoblotted with anti-Myc or anti-GFP, as indicated. The top panel shows the IP results. The middle panel represents the IP of GFP and GFP-Paf1. The bottom panel shows 3% input of the Myc fusion proteins used for each IP. The asterisk indicate the degradation of Myc-Ctr9.

-Figure S1: The nomenclature for heterodimers and fusion proteins is confusing.

"/" denotes protein complexes with separate chains (e.g., CTR9⁽¹⁻²⁴⁹⁾/PAF1⁽⁵⁷⁻¹¹⁶⁾), while "-" denotes heterodimers in a single-chain fusion (e.g., CTR9⁽¹⁻²⁴⁹⁾-PAF1⁽⁵⁷⁻¹¹⁶⁾). We have modified manuscript and legend accordingly.

-Line 96: change phrase to "eluted from size exclusion as a tripartite complex" or something similar

According to the reviewer's suggestion, we have revised the phrase to "eluted from the size-exclusion column as a tripartite complex".

-line 141 "TRP" should be "TPR"

Thank you for noticing this mistake. We have corrected this error in the revised manuscript.

-line 238 "Se-substituted" should be defined. It appears that the authors used Se-Met for phase determination, and this language should also be used in the text.

We have corrected this inconsistent description in the revised manuscript.

-line 411: the authors should state what the AH-1500 does. Is it a sonicator, French press, etc.

The AH-1500 is a type of high pressure cell cracker. We have included this information in the methods section.

-Line 430: It is unclear if the Ser-Gly additions are found at the N- or C- terminal side of the TEV site

DNA fragments were amplified by PCR and linked with a tobacco etch virus (TEV) protease-cleavable segment (Glu-Asn-Leu-Tyr-Phe-Gln-Ser). Two amino acids (Ser-Gly) were inserted both sides of the TEV protease-cleavable segment. We have corrected this unclear description in the methods section.

-Line 457: state as “the silver bullet additive” instead of “the silver bullet” and state manufacturer

Following the reviewer's suggestion, we have modified the manuscript accordingly.

-Line 494: °C appears to have been replaced by a box in a number of positions.

We have corrected the error accordingly in the revised manuscript.

Reviewer #2 (Remarks to the Author):

This manuscript describes a structure-function analysis of the association between the Paf1 and Ctr9 subunits of the Paf1 complex (Paf1C). Paf1C plays an important, albeit not fully understood, role in transcription elongation and connecting elongation to chromatin modifications, particularly activating marks on histone tails. Paf1C is critical for viability in yeast and has important connections with human disease. For example, some of the subunits and their interactions are thought to have tumor suppressor activity. There are few structural details about how the complex assembles and how the complex interacts with transcriptional machinery.

The authors have solved novel high-resolution crystal structures of two similar complexes, from human and yeast, containing the primary interacting regions of Paf1 and Ctr9. They compliment the structure with biochemical interaction studies in human cells and functional studies in yeast that exploit structure-based mutations. The main significant findings are the structural details of the Paf1-Ctr9 association, the mapping of several cancer-derived mutations to the interface, and the functional studies that demonstrate the Paf1-Ctr9 interface is critical for Paf1C assembly and function. The technical quality of the crystal structures is high and conclusions are properly supported by the complimentary biochemical data. The

study is for the most part rigorously done and well explained. Considering the importance of the complex and lack of structural details currently available, the work will have impact. It should be published after the authors are given the opportunity to consider the following minor concerns and suggestions:

We thank the reviewer for nicely summarizing the key contributions of our work, and for the constructive critiques and suggestions below.

1) It would be useful if the authors could explain more how they envision this subcomplex fitting into the full Paf1C complex. Although the previously published low resolution EM map may not allow for unambiguous docking of the high resolution structure, the authors could at least comment on different relationships between the subunits within the complex and perhaps show a schematic model. For example, the Ctr9 binding site in Paf1 is near the Leo1 binding site. Does Leo1 also contact Ctr9? More broadly, how is the observation that this local interaction is required for overall complex assembly consistent with a model for architecture?

We appreciate the reviewer pointing out some very important concern out. During revision, we have repeated co-IP experiments about yeast Paf1C assembly (revised Fig. 5a and 5b, and Fig. II, up). According to the newly acquired results, together with other evidence shown in previous studies (Ref. #45 and #46 in this manuscript), we have proposed a model for the assembly of yeast Paf1C (Fig. IV, below for convenience). In this model, Leo1 subunit binds to Paf1C through Paf1 subunit and the Ctr9/Paf1 heterodimer is the core component for Paf1C assembly. We have included Fig. IV as the revised Fig. 5c and modified the manuscript accordingly.

Fig. IV. Model of yeast Paf1C assembly. The Ctr9/Paf1 heterodimer is the core component for Paf1C assembly. The bold line represents the interaction in the crystal structure of the Ctr9⁽¹⁻³¹³⁾-Paf1⁽³⁴⁻¹⁰³⁾ heterodimer or the interaction between Paf1 and Leo1 obtained from the IP results and previous studies (ref. ^{45,46}). The fine lines represent the interaction obtained from the IP results. The dotted lines represent interactions that need to be further studied.

2) Line 84: The Ctr9-Caf1 heterodimer is called an “obligate” heterodimer, but it is not made clear what evidence supports this description. Was this published elsewhere? Fig S6 suggests that Ctr9 is aggregated in the absence of Paf1 association, but this part of the observation is not given any significant attention.

We have deleted the word "obligate" in the revised manuscript. Agree the reviewer's comments, the purified human CTR9⁽¹⁻²⁴⁹⁾ recombinant protein was aggregated in size-

exclusion column (blue line in **Fig. V**, below), comparing to CTR9⁽¹⁻²⁴⁹⁾/PAF1⁽⁵⁷⁻¹¹⁶⁾ complex protein (black line in **Fig. V**). Additionally, we found that yeast Ctr9⁽¹⁻³¹³⁾ forms inclusion bodies when expressing in *E. coli* (data now shown). These data indicate that both human CTR9/PAF1 and yeast Ctr9/Paf1 subcomplexes formation are essential for the folding of CTR9 and Ctr9 (at least for CTR9⁽¹⁻²⁴⁹⁾ and Ctr9⁽¹⁻³¹³⁾), respectively. We have included **Fig. V** as the revised **Supplementary Fig. 6a** and **a1-a3** and modified the sections of results and discussion accordingly.

Fig. V. CTR9⁽¹⁻²⁴⁹⁾ is aggregated in the absence of PAF1⁽⁵⁷⁻¹¹⁶⁾ binding. Analytical gel filtration profiles and SDS-PAGE of CTR9⁽¹⁻²⁴⁹⁾/PAF1⁽⁵⁷⁻¹¹⁶⁾ (black line in a and a1), CTR9⁽¹⁻²⁴⁹⁾/PAF1⁽⁵⁷⁻¹¹⁶⁾(5M) (dotted black line in a and a2), and CTR9⁽¹⁻²⁴⁹⁾ (blue line in a and a3). Those fractions shown in correspondent gels are indicated by a two-way arrow.

3) Lines 105-106: 1:1 stoichiometry cannot be rigorously determined from size-exclusion column (SE), particularly without standards. However, the 1:1 stoichiometry is convincingly shown by the molecular weight measurement using analytical ultracentrifugation data (and crystal structure). It would be more justified to change the conclusion from the SEC data to focus on the co-elution and the fact that the heterodimer and fusion elute in the same place, which supports a similar folded structure.

We fully agree with the reviewer on these comments. We have modified the manuscript accordingly.

4) Line 141: Change “five TRP” to “five TPR”.

Thank you for noticing this mistake. We have corrected this error in the revised manuscript.

5) It is curious that the A helix in Paf1 does not make contact with Ctr9 or anything that is clear in Fig. 2, yet it is ordered and somehow stabilized as a helix. Furthermore, there are some highly conserved residues and a disease related mutation in this helix. Does this helix contact a different Ctr9 in the crystal lattice that is related by symmetry? If so, this begs the

questions as to whether it could also make the equivalent interaction in solution. Does deletion of the A helix not change heterodimer stability? Do the authors have other thoughts on the role of this helix?

We thank the reviewer for pointing this very important concern out. We have analyzed two structures and found that both αA of PAF1⁽⁵⁷⁻¹¹⁶⁾ and $\alpha A'$ of Paf1⁽³⁴⁻¹⁰³⁾ do not contact with a different CTR9⁽¹⁻²⁴⁹⁾ and Ctr9⁽¹⁻³¹³⁾, respectively, in the crystal lattice. To further evaluate the role of this N-terminal helix, a truncated fragment of human PAF1 (aa 75-116, PAF1⁽⁷⁵⁻¹¹⁶⁾) or a truncated fragment of yeast Paf1 (aa 59-103, Paf1⁽⁵⁹⁻¹⁰³⁾) was designed to delete the N-terminal helix. As expected, size-exclusion chromatography and SDS-PAGE analysis showed the truncated PAF1⁽⁷⁵⁻¹¹⁶⁾ still can form a complex with CTR9⁽¹⁻²⁴⁹⁾ (Fig. VIa, below, insert), which is consistent with the observation that αA of PAF1⁽⁵⁷⁻¹¹⁶⁾ is not important for complex formation of CTR9⁽¹⁻²⁴⁹⁾-PAF1⁽⁵⁷⁻¹¹⁶⁾ (the revised Figs. 1 and 2). However, a temperature-dependent denaturation assay demonstrated that the truncated CTR9⁽¹⁻²⁴⁹⁾/PAF1⁽⁷⁵⁻¹¹⁶⁾ complex is less stable than CTR9⁽¹⁻²⁴⁹⁾/PAF1⁽⁵⁷⁻¹¹⁶⁾ complex (Fig. VIb, below), indicating that the αA of PAF1⁽⁵⁷⁻¹¹⁶⁾ may contribute to the stability of the CTR9⁽¹⁻²⁴⁹⁾/PAF1⁽⁵⁷⁻¹¹⁶⁾ complex. We have gotten similar conclusion about the role of $\alpha A'$ of Paf1⁽³⁴⁻¹⁰³⁾ in yeast Ctr9⁽¹⁻³¹³⁾/Paf1⁽³⁴⁻¹⁰³⁾ complex formation, according the results shown in the revised Supplementary Fig. 7b, b2 and 7d. We have included Fig. VI as the revised Supplementary Fig. 1h and 1i and modified the manuscript accordingly.

Fig. VI. (a) Analytical gel filtration profile and SDS-PAGE of CTR9⁽¹⁻²⁴⁹⁾/PAF1⁽⁷⁵⁻¹¹⁶⁾. (b) Differential scanning fluorimetry-based thermal denaturation assay showing the temperature-dependent denaturation profiles of CTR9⁽¹⁻²⁴⁹⁾/PAF1⁽⁷⁵⁻¹¹⁶⁾ and CTR9⁽¹⁻²⁴⁹⁾/PAF1⁽⁵⁷⁻¹¹⁶⁾ WT or mutants.

6) The authors could comment on the role of some of the most conserved residues in Paf1 Region I that do not appear to make contact with Ctr9 (D79, G81, and D85). For example, it appears in the view in Fig. S5A that G81 induces a bend that is critical for both L80 and V82 to fit into their respective hydrophobic pockets.

Thanks for the reviewer's suggestion. Structural-based alignment indicated that the residues D79, G81, and D85 in Region I of human PAF1 are conserved from yeast to human or the residue D95 in Region II are conserved from worm to human (**Fig. 2e**), indicating these residues have important roles in complexes formation. It was noted that G81 (cognate residue G63 of yeast Paf1) induces a bend that is critical for L80 and V82 (cognate residues L62 and M64 of Paf1) to fit into their respective hydrophobic pockets, respectively (revised **Supplementary Fig. 5a, d, h, and i**). Interestingly, the residues D79, D85, and D95 of PAF1 do not make contact with CTR9 (revised **Fig. 2b and 2c**), which is different to the cognate residues D61 and D67 involved in binding to Ctr9 (**Fig. VIIa**). However, these amino acids are very important for maintaining the local folding of PAF1. The side chain oxygen of D79 forms a hydrogen bond with the backbone nitrogen of G81; the side chain oxygen of D85 forms hydrogen bonds with the side chain hydroxyl oxygen of T91 and backbone nitrogen of I87 or N88; the side chain oxygen of D95 forms hydrogen bond with backbone nitrogen of N97 (**Fig. VIIb**, below). So we hypothesized that the disease-associated mutation D85N or D95Y will affect these hydrogen bonds and change the local folding of PAF1 (revised **Fig. 2g**). To this end, we tested the mutants PAF1⁽⁵⁷⁻¹¹⁶⁾(D85N) and PAF1⁽⁵⁷⁻¹¹⁶⁾(D95Y) in their binding to CTR9. Size-exclusion chromatography and SDS-PAGE analysis showed the mutant PAF1⁽⁵⁷⁻¹¹⁶⁾(D85N) or PAF1⁽⁵⁷⁻¹¹⁶⁾(D95Y) can form a complex with CTR9⁽¹⁻²⁴⁹⁾ (**Fig. VIIc-c2**, below). However, a temperature-dependent denaturation assay demonstrated that both the mutant complexes CTR9⁽¹⁻²⁴⁹⁾/PAF1⁽⁵⁷⁻¹¹⁶⁾(D85N) and CTR9⁽¹⁻²⁴⁹⁾/PAF1⁽⁵⁷⁻¹¹⁶⁾(D95Y) are less stable than CTR9⁽¹⁻²⁴⁹⁾/PAF1⁽⁵⁷⁻¹¹⁶⁾ WT complex (orange and magenta lines comparing to red line in **Fig. VIb**, up). We have included **Fig. VIIa-b** as the revised **Supplementary Fig. 5h-i** and modified the manuscript accordingly.

Fig. VII. The highly conserved residues are important for yeast Ctr9/Paf1 (**a**) and human CTR9/PAF1 (**b**) complexes formation. For clarify, only residues involved in hydrogen bonding are shown in (a) and (b), and the side chain are drawn in the stick model. Hydrogen bonding interactions are highlighted by dashed grey lines. Analytical gel filtration profiles and SDS-PAGE of CTR9⁽¹⁻²⁴⁹⁾/PAF1⁽⁵⁷⁻¹¹⁶⁾(D85N) (**c** and **c1**) and CTR9⁽¹⁻²⁴⁹⁾/PAF1⁽⁵⁷⁻¹¹⁶⁾(D95Y) (**c** and **c2**).

7) If possible, it would be a more impactful experiment if the co-IP in HEK cells were performed with full-length proteins. It is not surprising that the structure-derived mutations show the predicted effects when using identical constructs to those in the crystals. The more important question is whether the structural results are applicable in the context of the full-length proteins.

We agree the reviewer on this important point. During mapping the minimal binding region of interaction between human CTR9 and PAF1, we have designed a mutant lacking the N-terminal amino acids 1-116 (referred to as PAF1(N116 Δ)) and showed that the mutant GFP-[PAF1(N116 Δ)] did not co-immunoprecipitate with Myc-CTR9 thoroughly (lane 3 in **Fig. VIII**, below), indicating that the N-terminal fragment including amino acids 57-116 (aa 57-116 used in this study for structure determination) is required for the interaction between full-length CTR9 and PAF1. However, we found that CTR9 lacking amino acids 1-249 (aa 1-249 used in this study for structure determination) was not expressed in HEK293T cells.

Our previous study indicated that LEO1 binds to human PAF1C through PAF1 and that the CTR9 subunit is the key scaffold protein in assembling PAF1C (Ref. #46 in this manuscript). In this study, we took yeast Paf1C as example to show that Leo1 binds to Paf1C also through Paf1 and further confirm that the Ctr9/Paf1 is the core component for Paf1C assembly and functional regulation. Although these conserved structure and function, there exist some difference between human PAF1C and yeast Paf1C. For examples, human RTF1 is a less stable subunit of PAF1C (Refs. #5, #7, and #58 in this manuscript), while the cognate yeast Rtf1 binds to Paf1C tightly (lanes 2 in **Fig. 5a** and **5b**); and the observation that the N-terminal tail of yeast Ctr9 (not human CTR9) is essential for its binding to Paf1 (revised **Fig. 4**). Further studies are necessary to reveal the evolutionary structure and function or difference of this holo-complex. Over the past several years, we have tried very hard to obtain the holo-complex protein suitable for structural characterization and made some progress. Nevertheless, the *in vitro* and *in vivo* data presented in this manuscript clearly indicated that Ctr9/Paf1 subcomplex is the core component and is essential for holo-complex assembly. We have included **Fig. VIII** as the revised **Supplementary Fig. 1g** and modified the manuscript accordingly.

Fig. VIII. The N-terminal fragment amino acids 1-116 of PAF1 is required for the interaction between full-length CTR9 and PAF1. Extracts were prepared from HEK293T cells transfected with various combinations of plasmids, as indicated. The bottom panel shows 3% of the Myc fusion proteins for each IP.

Reviewer #3 (Remarks to the Author):

In the manuscript entitled “Paf1/Ctr9 subcomplex formation is essential for Paf1 complex assembly and functional regulation” Xie et al. present the structure of a subcomplex of the human and yeast Ctr9/Paf1. This subcomplex is part of the polymerase-associated factor 1 complex (PAF1C), which is involved in transcriptional elongation and chromatin regulation. There is limited structural information about PAF1C. This work builds up on previous work (Chu et al., 2013; Xu et al., 2017) and constitutes a step forward in our understanding of the complex formation at the molecular level. Furthermore, by using a combination of biophysical and biochemistry methods, the authors demonstrate that Ctr9/Paf1 is required for efficient assembly of PAF1C. In addition to the new structural data, the authors perform key mutations on the interface of Ctr9/Paf1 and monitor complex assembly and functionality in vivo assays. These point mutations confirm the impact that the correct assembly of PAF1C might have on cancer progression, since some of the previously identified mutations in cancer disease play also a key role in complex formation.

Overall, the science, methods and statistics presented in this work appear sound. There are no major issues, however some further clarification is needed on the points given below. Additionally, the paper would be easier to understand if the presentation was improved, e.g. simplification of figures.

We thank the reviewer for nicely summarizing the key findings of our work, and for the constructive critiques and suggestions below, especially in improving figure presentation.

1. Quality of Figures 4A, 4B and 5E is poor. Replacement with higher resolution images is a necessity.

We have used higher resolution images in the revised **Figs. 4a, 4b, and 6e.**

2. The authors perform co-immunoprecipitations (co-IP) experiments to test the interaction between hCtr9 and hPaf157-116. In Figure 2F, these assays are shown. In light of the crystal structure (hCtr91-249 - hPaf157-116), an extra co-IP assay showing the result of the interaction between hCtr91-249 and hPaf157-116 would complete Figure 2F (similar to lane 2 in Figure 2H). Alternatively, the authors should mention Figures 2F and 2H together whenever they refer to this experiment.

Following the reviewer's suggestion, we have repeated the co-IP experiments (**Fig. IX**, below for convenience), including the results of the interaction between CTR9⁽¹⁻²⁴⁹⁾ and

PAF1⁽⁵⁷⁻¹¹⁶⁾. We have included the newly acquired **Fig. IX** as the revised **Fig. 2f** and modified the manuscript accordingly.

Fig. IX. Co-IP experiments testing the interaction between CTR9⁽¹⁻²⁴⁹⁾ and PAF1⁽⁵⁷⁻¹¹⁶⁾ wild-type (WT) or mutants. The PAF1⁽⁵⁷⁻¹¹⁶⁾(4S) mutant contains four amino acid substitutions L80S, V82S, I84S, and L86S. The PAF1⁽⁵⁷⁻¹¹⁶⁾(5M) mutant contains five amino acid substitutions L80S, V82S, I84S, L86S, and D104K. Myc was tagged to the N-terminal of CTR9⁽¹⁻²⁴⁹⁾ WT or mutant and GFP was tagged to the C-terminal of PAF1⁽⁵⁷⁻¹¹⁶⁾ WT or mutant. Extracts were prepared from HEK293T cells transfected with various combinations of plasmids, as indicated. The bottom panel shows 3% of the Myc fusion proteins for each IP.

3. The authors attempt to define a new mode of interaction among TPR motifs. TPR motifs are widespread in nature and present a common fold. Further evidence would be needed to conclude that a novel TPR-SLIM (small linear motif) interaction has been found. This point seems over-interpreted. Comprehensive bioinformatics analysis and a figure showing a comparison of structural results (and different type of folds) would be necessary to raise this conclusion. Otherwise, the authors are encouraged to moderate (or even omit) this statement in the text.

We fully agree with the reviewer on this comment and have stated the phrase in section of abstract or discussion to "Here we report two structures of each of the human and yeast Paf1/Ctr9 subcomplexes, which assemble into heterodimers with very similar conformations, revealing a previously uncharacterized interface between the tetratricopeptide repeat (TPR) module in Ctr9 and Paf1."

4. To validate the interactions observed in the structure of hCtr91-249/hPaf157-116 a number of mutagenesis studies are presented. The overall conclusion of this study seems precise. However, Figure S6 and related comments need to be improved for clarity. For example, all mutants (except hCtr91-249 (R98W)/hPaf157-116, Figure S6C, blue lane) seem to aggregate. This is a typical behaviour of disrupted complexes that occurs when one or more of the interactor partners are missing. In Figures S6A and S6A2, Ctr9 seems to appear in the void volume (first peak of SEC, S6A) and Paf1 shows up later (second peak of SEC, S6A). In the same figure, it also seems like Ctr9 is also present in this second peak (Figure S6A2, lane 35) and shows the same intensity as Paf1. This does not correlate with the absorbance of the

chromatogram peaks. In addition, in S6A2 there are 2 bands of similar size (14.4 kDa) and it is unclear which one corresponds to hPaf157-116(5M). Could the authors elaborate further on the interpretation of the chromatograms and corresponding gels, and specifically on Figures S6A and S6A2? Could it be that mutated complexes are still observed in the second peak of the chromatograms (~fraction 35) but with disrupted stoichiometry? See comments on how to improve Figure S6A below.

We fully agree the reviewer on these comments and appreciate the reviewer pointing out some very important concern out. During revision, we have purified CTR9⁽¹⁻²⁴⁹⁾ and CTR9⁽¹⁻²⁴⁹⁾/PAF1⁽⁵⁷⁻¹¹⁶⁾(5M). Size-exclusion chromatography and SDS-PAGE analysis showed that the purified CTR9⁽¹⁻²⁴⁹⁾ is aggregated in the absence of PAF1⁽⁵⁷⁻¹¹⁶⁾ binding (dotted line and blue line in Fig. X, below). Additionally, we found that yeast Ctr9⁽¹⁻³¹³⁾ forms inclusion bodies when expressing in *E. coli* (data now shown). These data indicate that both human CTR9/PAF1 and yeast Ctr9/Paf1 subcomplexes formation are essential for the folding of CTR9 and Ctr9 (at least for CTR9⁽¹⁻²⁴⁹⁾ and Ctr9⁽¹⁻³¹³⁾), respectively. We have included Fig. X as the revised Supplementary Fig. 6a and a1-a3 and modified the sections of results and discussion accordingly.

Fig. X. CTR9⁽¹⁻²⁴⁹⁾ is aggregated in the absence of PAF1⁽⁵⁷⁻¹¹⁶⁾ binding. Analytical gel filtration profiles and SDS-PAGE of CTR9⁽¹⁻²⁴⁹⁾/PAF1⁽⁵⁷⁻¹¹⁶⁾ (black line in a and a1), CTR9⁽¹⁻²⁴⁹⁾/PAF1⁽⁵⁷⁻¹¹⁶⁾(5M) (dotted black line in a and a2), and CTR9⁽¹⁻²⁴⁹⁾ (blue line in a and a3). those fractions shown in correspondent gels are indicated by a two-way arrow.

5. Lines 85-86 “elucidated the mechanism responsible for human diseases caused by mutations of Paf1 and Ctr9” are an overstatement of the conclusions. The authors have demonstrated that the assembly of PAF1C might have an impact on specific cancer diseases, but further investigation is needed to elucidate the mechanism responsible for the human disease.

We agree the reviewer for this point. We have removed this statement and modified the manuscript accordingly.

Suggestions to improve figures.

Figure 1A.

- Summarizing disease-associated mutations on this figure is not constructive. A table showing a summary of mutations is recommended. For simplicity, information about mutations should be omitted in the figure. This way, this figure would compare easily to Figure 3A.

Instead, the schematic representation should show all domains present in hCtr9 and hPaf1 (full domain representation) for clarity. Schematic representation of Figure 3A should be modified accordingly.

We fully agree with the reviewer on this point. According to the reviewer's suggestion, we have omitted the mutations information in the original Fig. 1A and shown all domains in CTR9 and PAF1 in **Figure XI** below. Besides, the full lists of the disease-associated mutations of CTR9 and PAF1 were summarized in **Table I** and **Table II** below, respectively. We have included the **Figure XI** as revised **Fig. 1a** and the **Table I** and **Table II** as the revised **Supplementary Table 2** and **Table 3**, respectively.

Fig. XI. Schematic representation of full-length CTR9 and PAF1. A LEO1-interacting region (blue) and a histone H3-interacting region (orange) are shown in PAF1. The predicted 19 TPR motifs (gray) were defined using TPRpred (<https://toolkit.tuebingen.mpg.de/#tools/tprpred>). The region fragments of the CTR9⁽¹⁻²⁴⁹⁾/PAF1⁽⁵⁷⁻¹¹⁶⁾ complex used for structural determination are indicated by a two-way arrow and are colored cyan and magenta, respectively.

Table I. Disease-associated mutations in human CTR9

Mutation type	Mutation (Amino acid)
Missence	L10F, E15G, L19I, P25L, Q36R, I43V, A46D, A48V, L63V, A67T, D74H, D77V, L89F, R98W, N102K, D108V, T118A, D121N, Y126C, N129I, H153Y, N157S, S159F, P164S, K169N, S173F, Y185C, K188N, R191H, A198T, R201C, K213T, E215K, A220T, A224S, E226K, C231F, V232M, S250F, S267N, L272W, A276E, V288F, H294Y, M304K, E320G, Y329C, F344C, E376K, G382S, A386T, G399C, H400Y, K402E, D426N, A437T, P450T, L454F, N456D,

	A459V, G468W, R480S, R480C, A494T, A494S, A494P, A494V, A509V, R525L, E526K, Y534C, R536H, E556A, W568L, G572D, G572V, N573H, W581C, P583H, S596F, G608S, Q613K, R620Q, K624N, R627H, R631H, R631C, K637R, R641K, K645M, N646K, L657V, H659P, R664C, R667C, T677S, A678T, D682V, A688V, A700T, Y704C, Y704D, H714D, V720I, Y722C, A747T, D750N, L764F, D771N, A782V, A789V, Y792N, L796F, V799A, V799L, G800E, D806H, S818F, L820V, R830Q, R830W, R832H, D835Y, E838Q, R839Q, R839W, E846Q, K849N, R853M, L857P, Q860H, R864H, K872N, L874P, R878Q, R878W, E883K, L889I, M890T, F891C, R903G, R910H, R910C, S911C, S911F, E916D, D924Y, D926G, E948D, R961I, G966R, K982Q, R984Q, R985C, P986T, K993R, R999C, P1002L, S1003L, K1007R, S1015L, D1018H, D1022N, L1026H, K1027R, S1059F, N1062S, G1067D, E1069K, G1071S, R1075K, R1099Q, R1080W, P1090S, S1097Y, E1105K, R1122H, S1143Y, N1157I, S1161L, S1165L, D1173N
Nonsense	G27*, Q391*, R394*, E515*, E526*, W553*, Q629*, G654*, G800*, R1077*
Frameshift deletion	T57fs*5, K107fs*96, L475fs*63, V673fs*13, R903fs*102, D923fs*1
Frameshift insertion	L24fs*6 , L368Ffs*6
In-frame deletion	R962delR

Various disease-associated mutations in CTR9 were listed, based on the data extracted from the COSMIC database (<http://cancer.sanger.ac.uk/cosmic>). A total of 38 missense mutations were located in 38 residues of CTR9⁽¹⁻²⁴⁹⁾. These 38 mutations were divided into three categories. The three categories were named as interface, folding, and others, and amino acid substitutions in each category were colored in red, orange, and green, respectively. Only mutations used in this study were shown in **Fig. 2g**.

Table II. Disease-associated mutations in human PAF1

Mutation type	Mutation (Amino acid)
---------------	-----------------------

Missence	R11W, R11L, S38I, D54N, H70Q, K71R, H72N, H72R, H72Q, D73H, D73V, L74I, L74R, T76S, D85N, L86F, I87M, D95N, D95Y, E105D, E109K, A125T, K133N, T134I, E149K, E167K, I176L, P201L, E203D, M250V, D251N, G254R, F261S, E266K, T267M, R271Q, R273Q, E277K, D285N, W297L, N301H, G306D, G318C, R331C, R335H, R336Q, S343L, A347T, R354W, M356T, E358K, K359T, R367W, R367Q, P376L, E412Q, R420W, R420Q, K431R, K431N, E438K, E441K, R445Q, E469K, D470N, R471S, Q475L, D481A, G489A, R492W, S495N, A519V, D522N, A526D, D527N, D527H, D531N
Nonsense	E25*, G28*, W130*, R132*, R198*, E469*, G472*
Frameshift deletion	T76fs*24, S174fs*48, K230fs*11, Q475fs*>57, G485fs*>47
Frameshift insertion	G401fs*>134, G408fs*>125
In-frame deletion	F313delF, E399delE, K406delK, A519_A526delAASDSS...

Various disease-associated mutations in PAF1 were listed, based on the data extracted from the COSMIC database. A total of 21 missense mutations were located in 15 residues of PAF1⁽⁵⁷⁻¹¹⁶⁾. These 21 mutations were divided into three categories. The three categories were named as interface, folding, and others, and amino acid substitutions in each category were colored in red, orange, and green, respectively. Only mutations used in this study were shown in **Fig. 2g**.

Figures 1B-D.

- Labels for hCtr91-249 and hPaf157-116 should be added to the cartoon representations for easy interpretation of the figure.
- It would be easier to interpret the structural panels if they all show the protein structure in a similar manner, eg. cylinders vs alpha helices. This would make it easier to see how the various views are related to each other.
- The same applies to Figures 3B-C.

According to the reviewer's suggestion, helices in protein structure were shown in cylinders and the color codes CTR9⁽¹⁻²⁴⁹⁾ in cyan, and PAF1⁽⁵⁷⁻¹¹⁶⁾ in magenta or Ctr9⁽¹⁻³¹³⁾ in orange and Paf1⁽³⁴⁻¹⁰³⁾ in green were added in the revised **Fig. 1b** and **Fig. 2a**, or **Fig. 3b**, respectively.

Figure 2.

- Figures 2B, 2C and 2D. Interpretation of these figures is difficult. A LigPlot or a cartoon representation with a zoom of the highlighted areas showing the residues involved in the interactions would clarify the figure considerably. The same applies to Figures 3D-F.

Thanks for suggestion. Following the reviewer's suggestion, in the revised **Fig. 2e-d** and **Fig. 3e-g**, we have used LigPlot to show the residues involved in the interactions between human CTR9 and PAF1 (**Fig. XII**, below) heterodimer or yeast Ctr9 and Paf1 heterodimer (**Fig. XIII**, below), respectively. Alternatively, in the revised **Supplementary Fig. 5a-f**, we have shown the interaction details between CTR9 and PAF1 or Ctr9 and Paf1 with cartoon representation in stereo view.

Fig. XII. The interaction details between CTR9 and PAF1 in the three regions are shown in (b) to (d). Charge-charge or hydrogen-bonding and hydrophobic interactions are shown as gray dotted lines and spoked arcs, respectively.

Fig. XIII. The interaction details between yeast Ctr9 and Paf1 in the three regions are shown in (e) to (g). Charge-charge or hydrogen-bonding and hydrophobic interactions are shown as gray dotted lines and spoked arcs, respectively.

- Figure 2E. The name of the proteins used to depict secondary structures on top (hPaf1) and bottom (yPaf1) of alignment should be indicated for clarity. Have these secondary structures been predicted with any specific program? The same applies to Figure S2 (hCtr9 and yCtr9).

We have added the protein names at the top and bottom of each alignment shown in **Fig. 2e** and **Supplementary Fig. 2**. All sequence alignments shown in this manuscript are based the crystal structures of CTR9⁽¹⁻²⁴⁹⁾-PAF1⁽⁵⁷⁻¹¹⁶⁾ and Ctr9⁽¹⁻³¹³⁾-Paf1⁽³⁴⁻¹⁰³⁾, which are determined in this study.

- Figures 2F and 2H. Labels of specific band proteins are not shown in Figures 2F and 2H. Also, is hPaf157-116 tagged with GFP in N or C terminus? (N-terminal residue in Figure 2F and C-terminal one in Figure 2H).

Thanks for pointing out the mistakes. Myc was tagged to the N-terminal of CTR9⁽¹⁻²⁴⁹⁾ and GFP was tagged to the C-terminal of PAF1⁽⁵⁷⁻¹¹⁶⁾ WT or mutant. For clarify, we have stated the position of each tag in all constructs used in Co-IP experiments.

- Figure 2G. To make it easier understand what type or residues are included in categories 1, 2, 3 (both mentioned in the text and in this figure), these could be renamed as ‘interface’,

‘folding’ and ‘others’, respectively. This figure is difficult to interpret and it is not very informative (again a comprehensive table with mutations would help if added). To make this figure more useful, the authors could focus on showing just the mutations used in this work (categories 1 and 3).

Thanks for suggestion. Following the reviewer's suggestion, we have revised **Fig. 2g** (**Fig. XIV**, below) and modified the manuscript accordingly.

Fig. XIV. Disease-associated mutations in the CTR9⁽¹⁻¹²⁹⁾/PAF1⁽⁵⁷⁻¹¹⁶⁾ complex. For clarify, only five missense-mutation sites in category 1 (interface) of PAF1, and two sites in category 2 (folding) of PAF1, and one site (R98W) in category 3 (others) of CTR9 are highlighted with spheres and colored in red, orange, and green, respectively. The full lists of disease-associated mutations in CTR9 and PAF1 are summarized in Supplementary Table 2 and 3 (**Table I** and **II**, up), respectively.

Figure 5E. Label of IB is missing.

We have improved this unclear presentation in the revised **Fig. 6e**.

Figure S1.

- Contrast of S1B needs to be improved.
- Color code for chromatograms is not shown in S1E.

In the revised **Supplementary Fig. 1**, the original SDS-PAGE in (b) was replaced by a new one and the color codes in (e) were added.

Figure S3.

This figure can be used to highlight the differences in the N-terminal domain of the human and yeast proteins, which will complement Figure 3H. This represents an important finding in this paper, i.e. this N-terminal region is essential to maintain the interaction between yCtr9 and yPaf1.

Thanks for reviewer's suggestion. Structural-based sequence alignment of the N-terminal tail of Ctr9 in various species indicated that yeast Ctr9 has a longer N-terminal tail than other species (e.g., human CTR9) (Fig. XVa, below). The Ctr9⁽¹⁻³¹³⁾-Paf1⁽³⁴⁻¹⁰³⁾ structure showed that residues Y10, P11, M13, E14, and W15 in the longer N-terminal tail of yeast Ctr9 are directly involved in binding to Paf1 (revised Fig. 3f and 3g). Fig. XVb clearly showed that the [Ctr9(N16Δ)] mutant, in which the most N-terminal 16 amino acids of Ctr9 were deleted, could not form complex with Paf1 thoroughly, indicating this longer N-terminal tail of Ctr9 is essential to maintain the interaction between Ctr9 and Paf1. We have included Fig. XV as the revised Fig.4 and modified the manuscript accordingly.

Fig. XV. The longer N-terminal tail of Ctr9 is essential for its binding to Paf1. (a) Structural-based sequence alignment of the N-terminal fragments of Ctr9 in various species. In this alignment, the secondary structures of human CTR9⁽¹⁻³⁸⁾ and yeast Ctr9⁽¹⁻⁵³⁾ are shown at the top and bottom, respectively, and conserved residues are shaded in red. The amino acids 1-16 of yeast Ctr9, which were deleted in the GFP-Ctr9(N16Δ) construct [used in (b)], are marked with a dotted blue box. The amino acids Y10, P11, M13, E14, and W15 of Ctr9 involved in its binding to Paf1 are marked by orange stars. (b) Co-IP experiments testing the interactions between Ctr9 WT or Ctr9(N16Δ) mutant and Paf1. Extracts were prepared from HEK293T cells transfected with various combinations of plasmids, as indicated. The bottom panel shows 3% of the Myc fusion proteins for each IP.

Figure S4C.

To appreciate the differences between human and yeast Ctr9/Paf1, the use of only two colours would simplify the representation.

Following the reviewer's suggestion, human CTR9/PAF1 is colored in cyan and yeast Ctr9/Paf1 is colored in orange (revised Supplementary Fig. 4c)

Figure S5.

- S5A and S5B. It is not possible to appreciate the stick representation of residues very well. A zoom highlighting these regions would help the reader. Labels in green and blue are not easy to read. A possibility could be to use lighter colours.

We agree the reviewer on this point. We have removed the original S5A and S5B. In the revised Supplementary Fig. 5a-f, we have shown the interaction details between CTR9 and PAF1 or between Ctr9 and Paf1 with cartoon representation in stereo view. Following the reviewer's suggestion, in the revised Fig. 2e-d and Fig. 3e-g, we have used LigPlot to show

the residues involved in the interactions between human CTR9 and PAF1 or yeast Ctr9 and Paf1 heterodimer, respectively.

- Figure S5E. Not mention in the text.

The revised **Supplementary Fig. 7g** is mentioned in the phrase of "Our structural analyses show that both ligands human PAF1 and yeast Paf1 formed a similar hook-fold" .

Figure S6.

- Figures S6A, B, C. Label for “fractions” in chromatograms as well as asterisks on top of those fractions shown in correspondent gels are necessary to clarify figure.
- Gels. Represent fraction number in just one lane would help reading of the figure.

Thanks for reviewer's suggestion. In the revised **Supplementary Fig .6**, fractions are labeled in each gel filtration profiles; those fractions shown in correspondent gels are indicated by a two-way arrow; and represent fraction number are wrote in one lane.

Minor points.

- Several parts of the manuscript need to be clarified so it is more suitable for a general audience. This will aid the reader understand better the text. For example, a general overview of what it is known about Paf1, Ctr9 and PAF1C from the structural point of view is missing.

Following the reviewer's suggestions. We have included the structural information about Paf1 complex in the revised introduction. The structures of Ras-like domain of Cdc73 (Refs. #38 and #39 in this manuscript) and the Plus3 domain of RTF1 (Refs. #40, #41, and #42 in this manuscript) provide the structural basis for Paf1 complex chromatin association. Recently, the crystal structure of the N-terminal domain of CDC73 has been resolved, which may provide the molecular mechanisms of hyperparathyroidism-jaw tumor mutants (Ref. #43 in this manuscript). The structure of histone modification domain (HMD) of Rtf1 was reported and the HMD was shown to stimulate H2B ubiquitylation through interaction with Rad6 (Ref. #44 in this manuscript). However, there is no atomic structure information has been reported about Ctr9, which contains multi-motifs for protein-protein interaction (Figs. 1a and 3a).

- ‘Crystal structure of the hCtr9/hPaf1 heterodimer’ section. It is unclear how the TEV constructs have been made. Is the TEV site cleaved at all? Specify in line 117 where the TEV-cleavable segment has been included. Explain how the artificial linker is helping crystallization.

Thanks for reviewer for pointing these concern out. We have fused CTR9⁽¹⁻²⁴⁹⁾ to the N-terminus of PAF1⁽⁵⁷⁻¹¹⁶⁾ with a TEV-cleavable segment to make a single-chain fusion protein of the CTR9⁽¹⁻²⁴⁹⁾-PAF1⁽⁵⁷⁻¹¹⁶⁾ complex. DNA fragments were amplified by PCR and linked with a tobacco etch virus (TEV) protease-cleavable segment (Glu-Asn-Leu-Tyr-Phe-Gln-Ser). Two amino acids (Ser-Gly) were inserted both sides of the TEV protease-cleavable

segment. Similar strategy was applied to make single-chain fusion protein of Ctr9⁽¹⁻³¹³⁾-Paf1⁽³⁴⁻¹⁰³⁾ complex. During crystallization, we did not add any protease to proteins sample and thus SDS-PAGE analysis of dissolved crystals demonstrated that the molecules were intact (revised **Supplementary Fig. 3b** and **3d**). It is possible that the artificial linker may stabilize the N-terminus of PAF1⁽⁵⁷⁻¹¹⁶⁾ or Paf1⁽³⁴⁻¹⁰³⁾ and thus helping complex proteins crystallization. We have modified the manuscript accordingly.

- Lane 267. Elaborate further on the interaction between the N-terminal tail of yCtr9 and yPaf1 and how this is different to the human system.

Following the reviewer's suggestion, we have included a newly **Fig. 4a**, indicating that yeast Ctr9 has a longer N-terminal tail than other species (e.g., human CTR9) (**Fig. XVa**, up). The Ctr9⁽¹⁻³¹³⁾-Paf1⁽³⁴⁻¹⁰³⁾ structure showed that residues Y10, P11, M13, E14, and W15 in the longer N-terminal tail of yeast Ctr9 are directly involved in binding to Paf1 (revised Fig. 3f and 3g). **Fig. XVb** clearly showed that the [Ctr9(N16Δ)] mutant, in which the most N-terminal 16 amino acids of Ctr9 were deleted, could not form complex with Paf1 thoroughly, indicating this longer N-terminal tail of Ctr9 is essential to maintain the interaction between Ctr9 and Paf1. We have modified the manuscript accordingly.

- Lane 268. Specify that yL83 is tested because it is only present in yeast.

Following the reviewer's suggestion, we have modified the manuscript accordingly.

- Different terms are used for co-immunoprecipitation (co-IP). For example in lines 290 and 294 it is named as coprecipitation. Check consistency for this term.

We have corrected this inconsistent terms in the revised manuscript.

- A cartoon showing the updated proposed assembly of PAF1C is recommended. This would enhance the result and discussion.

According to the structural and biochemical results in this study, together with other evidence shown in previous studies (Ref. #48 and 46 in this manuscript), we have proposed a model for the assembly of yeast Paf1C (**Fig. XVI**, below). We have included **Fig. XVI** as the revised **Fig. 5c** and modified the manuscript accordingly.

Fig. XVI. Model of yeast Paf1C assembly. The Ctr9/Paf1 heterodimer is the core component for Paf1C assembly. The bold line represents the interaction in the crystal structure of the Ctr9⁽¹⁻³¹³⁾-Paf1⁽³⁴⁻¹⁰³⁾ heterodimer or the interaction between Paf1 and Leo1 obtained from the IP results and previous study (ref.^{45,46}). The fine lines represent the interaction obtained from the IP results. The dotted lines represent interactions that need to be further studied.

- D95K mutation in yPaf1 abolished the interaction between yCtr9 and yPaf1 completely. It would be interesting to test if D104K in hPaf1 had the same effect.

Following the reviewer's suggestion, we have tested the ability of the mutant PAF1⁽⁵⁷⁻¹¹⁶⁾(D104K) binding to CTR9⁽¹⁻²⁴⁹⁾ in a co-IP assay. As expected, the C-terminal GFP-tagged mutant PAF1⁽⁵⁷⁻¹¹⁶⁾(D104K)-GFP did not co-immunoprecipitate with Myc-tagged CTR9⁽¹⁻²⁴⁹⁾ (Myc-CTR9⁽¹⁻²⁴⁹⁾) (lane 5 in **Fig. XVII**, below). We have included **Fig. XVII** as the revised **Fig. 2f** and modified the manuscript accordingly.

Fig. XVII. Co-IP experiments testing the interaction between CTR9⁽¹⁻²⁴⁹⁾ and PAF1⁽⁵⁷⁻¹¹⁶⁾ wild-type (WT) or mutants. The PAF1⁽⁵⁷⁻¹¹⁶⁾(4S) mutant contains four amino acid substitutions L80S, V82S, I84S, and L86S. The PAF1⁽⁵⁷⁻¹¹⁶⁾(5M) mutant contains five amino acid substitutions L80S, V82S, I84S, L86S, and D104K. Myc was tagged to the N-terminal of CTR9⁽¹⁻²⁴⁹⁾ WT or mutant and GFP was tagged to the C-terminal of PAF1⁽⁵⁷⁻¹¹⁶⁾ WT or mutant. Extracts were prepared from HEK293T cells transfected with various combinations of plasmids, as indicated. The bottom panel shows 3% of the Myc fusion proteins for each IP.

REVIEWERS' COMMENTS:

Reviewer #1 (Remarks to the Author):

The authors successfully addressed our concerns and the manuscript should be published. It is a valuable contribution to the field.

I only recommend to consider to rephrase:

" According to the sequence alignment of PAF1 from various species, we found a region in PAF1 (aa 57-266, PAF1(57-266)) that formed a heteromeric complex with CTR9(1-284) independent of the LEO1 subunit (Supplementary Fig. 1b)."

to:

We found that a conserved N-terminal fragment of PAF1 (57-266) formed a stable dimeric complex with CTR9 1-284 (supplementary Fig. 1b).

Reviewer #2 (Remarks to the Author):

The revised version of the manuscript addresses previous concerns adequately with additional experiments and clarification in the text. The manuscript is suitable for publication.

Reviewer #3 (Remarks to the Author):

This is an exciting study into the assembly of the eukaryotic polymerase-associated factor 1 complex (PAF1C and Paf1C). The authors describe thoroughly the importance of the Ctr9/Paf1 sub-complex in the formation of PAF1C and Paf1C, clarify the role of specific disease mutations and explore further their role in yeast.

The authors have addressed all of my comments in an excellent manner. I support publication of this manuscript in Nature Communications.

Notice the following:

- In line "Notably, GFP-tagged [PAF1(N116Δ)] [GFP-PAF1(N116Δ)] mutant did not co-immunoprecipitate", it should be immunoprecipitate
- Labels in ligplot (hydrogen bond distances), font too small
- Order of letter in figures would be better is shown from right to left, and up to bottom, whenever possible.

Point-by-point responses to the reviewers' comments:

(Our responses to the reviewers' comments are highlighted in blue)

Reviewer #1 (Remarks to the Author):

The authors successfully addressed our concerns and the manuscript should be published. It is a valuable contribution to the field.

We thank Reviewer #1 for his/her final approval.

I only recommend to consider to rephrase:

" According to the sequence alignment of PAF1 from various species, we found a region in PAF1 (aa 57-266, PAF1(57-266)) that formed a heteromeric complex with CTR9(1-284) independent of the LEO1 subunit (Supplementary Fig. 1b)."

to:

We found that a conserved N-terminal fragment of PAF1 (57-266) formed a stable dimeric complex with CTR9 1-284 (supplementary Fig. 1b).

Thanks for suggestion. Following the reviewer's suggestion, we have modified the manuscript accordingly.

Reviewer #2 (Remarks to the Author):

The revised version of the manuscript addresses previous concerns adequately with additional experiments and clarification in the text. The manuscript is suitable for publication.

We thank Reviewer #2 for his/her final approval.

Reviewer #3 (Remarks to the Author):

This is an exciting study into the assembly of the eukaryotic polymerase-associated factor 1 complex (PAF1C and Paf1C). The authors describe thoroughly the importance of the Ctr9/Paf1 sub-complex in the formation of PAF1C and Paf1C, clarify the role of specific disease mutations and explore further their role in yeast.

The authors have addressed all of my comments in an excellent manner. I support publication of this manuscript in Nature Communications.

We thank Reviewer #3 for his/her final approval.

Notice the following:

- In line "Notably, GFP-tagged [PAF1(N116Δ)] [GFP-PAF1(N116Δ)] mutant did not co-immunoprecipitate", it should be immunoprecipitate
- Labels in ligplot (hydrogen bond distances), font too small
- Order of letter in figures would be better is shown from right to left, and up to bottom, whenever possible.

Thanks for suggestion. According to the reviewer's notice, we have modified the manuscript and figures accordingly (e.g., the word co-immunoprecipitate was corrected to immunoprecipitate; font size was set up to 8 pt; the order of letter in figures was shown from left to right and up to down when it is possible).